# Periaqueductal gray neurons encode the sequential motor program in hunting behavior of mice

Hong Yu[1,2,3,8], Xinkuan Xiang[1,2,8], Zongming Chen[4,8], Xu Wang[1,2], Jiaqi Dai[1,2], Xinxin Wang[1,2], Pengcheng Huang[1,2], Zheng-dong Zhao[5], Wei L. Shen [4✉] & Haohong Li[1,2,6,7✉]

Sequential encoding of motor programs is essential for behavior generation. However, whether it is critical for instinctive behavior is still largely unknown. Mouse hunting behavior typically contains a sequential motor program, including the prey search, chase, attack, and consumption. Here, we reveal that the neuronal activity in the lateral periaqueductal gray (LPAG) follows a sequential pattern and is time-locked to different hunting actions. Optrode recordings and photoinhibition demonstrate that LPAG[Vgat] neurons are required for the prey detection, chase and attack, while LPAG[Vglut2] neurons are selectively required for the attack. Ablation of inputs that could trigger hunting, including the central amygdala, the lateral hypothalamus, and the zona incerta, interrupts the activity sequence pattern and substantially impairs hunting actions. Therefore, our findings reveal that periaqueductal gray neuronal ensembles encode the sequential hunting motor program, which might provide a framework for decoding complex instinctive behaviors.

[1] Britton Chance Center for Biomedical Photonics, Wuhan National Laboratory for Optoelectronics, Huazhong University of Science and Technology, Wuhan, Hubei 430074, China. [2] MoE Key Laboratory for Biomedical Photonics, Collaborative Innovation Center for Biomedical Engineering, School of Engineering Sciences, Huazhong University of Science and Technology, Wuhan, Hubei 430074, China. [3] College of Basic Medicine, Hubei University of Medicine, Shiyan, Hubei 442000, China. [4] School of Life Science and Technology and Shanghai Institute of Advanced Immunochemical Studies, Shanghaitech University, Shanghai 201210, China. [5] Program in Cellular and Molecular Medicine, Boston Children's Hospital, Boston, MA 02115, USA. [6] Affiliated Mental Health Centre and Hangzhou Seventh People's Hospital, Zhejiang University School of Medicine, Hangzhou, Zhejiang 310013, China. [7] The MOE Frontier Research Center of Brain & Brain-machine Integration, Zhejiang University School of Brain Science and Brain Medicine, Hangzhou, Zhejiang 310058, China. [8]These authors contributed equally: Hong Yu, Xinkuan Xiang, Zongming Chen. ✉email: shenwei@shanghaitech.edu.cn; hhli_27@zju.edu.cn

Predatory hunting is an evolutionarily conserved behavior across the animal kingdom, which is vital to the survival of organisms[1]. It is a well-established instinctive behavior with a specific sequential behavioral action, including the prey search, pursuit, attack, and consumption, which can be readily recapitulated in a laboratory setting[2–4]. These behavioral actions are very stereotypical under the same conditions, but flexible among treatments or alternative stimuli[5,6], making them highly suitable for addressing the fundamental question of how neuronal activity drives complex behavior. Further, since the hunting behavior is highly conserved, understanding its neural basis is likely to uncover the basic principles of behavioral organization. The sequential neuronal activity has been established in the hippocampus and motor cortex during memory or decision-making tasks[7–11], which is essential for navigation planning[7] and motor generation[11]. However, the neural substrate underlying instinctive behavioral sequences such as that in hunting is still mostly unknown.

Many brain regions have been associated with predatory hunting, including the superior colliculus[12,13], the amygdala[14], the hypothalamus[2], and the periaqueductal gray[3,15]. Interestingly, our previous work[13,16] and other recent studies[2,14] have found that optogenetic activation neurons in either the zona incerta (ZI), the central amygdala (CeA), or the lateral hypothalamus (LH) are sufficient to elicit a hunting-like behavior in prey-naive mice, suggesting that the areas transmit key signals to induce the hunting motor program. All these three areas, together with neurons in the medial preoptic area (MPA)[17], target neurons in the periaqueductal gray matter (PAG) to exert their function, suggesting that the PAG is a hub to control the hunting motor program.

The PAG is a midbrain structure with multiple functions, including defensive[18–20], social[21], maternal[22], and emotional behaviors[23]. From dorsal to ventral direction, the region is divided into four function-specific columns, including the dorsomedial (DMPAG), dorsolateral (DLPAG), lateral (LPAG), and ventrolateral (VLPAG) columns[24–26]. Lesions of the LPAG in mice impair their ability to chase or attack the prey[27]. However, how the LPAG integrates multiple inputs to guide hunting remains largely unknown. The hunting behavior is proceeding with a strict sequence, containing four different motor actions as aforementioned[5,6]. Therefore, like in the hippocampus and motor cortex, we hypothesized that LPAG neuronal activities are organized in a sequential framework to support the serial structure of hunting motor actions.

Here, by applying in vivo optrode recordings and genetically encoded circuit analysis tools, we demonstrated that a neuronal ensemble sequence was formed in the LPAG across the whole predatory process. We further identified that LPAG$^{Vgat}$ neurons were recruited to support the prey search, chase and attack, but not eating, while LPAG$^{Vglut2}$ neurons were only recruited to support the attack. By input-specific lesions and in vivo single-unit recordings, we found that the CeA, the LH, and the ZI convey distinct hunting-related information to the LPAG to regulate distinct phases of predatory hunting, but they all were required for the formation of the intact activity sequence. Together, our data reveal that LPAG neuronal ensembles encode sequential hunting motor actions and this sequential pattern is critically regulated by upstream input signals.

## Results

**Distinct clusters of LPAG neurons are sequentially recruited in hunting.** To investigate how LPAG neurons are engaged in predatory hunting, we employed in vivo single-unit recordings in freely moving mice during the cricket task (Fig. 1a). Predatory hunting could be divided into four phases according to previous studies[2,3,16], including the prey introduction, chase, attack, and eating (Fig. 1b). We have shown that in a small arena, cricket-trained mice will immediately detect the prey after introduction, reorient to chase the prey after a latency, and then attack the prey, with these behavioral actions repeating until successful prey capture[16]. Captured prey is often eaten in mice, although hunted prey might be saved for later eating in nature and the neural circuits for hunting and feeding are separable[14,16]. We confirmed the electrode locations in all recorded mice (Fig. 1c and Supplementary Fig. 1a), and a successful hunting trial usually contained multiple chase and attack phases (Supplementary Fig. 2a–c). We recorded the activity of LPAG neurons ($n = 618$ units, ten mice) in C57BL/6J mice and compared the neuronal activity in hunting phases to a distribution generated by shuffling the neuronal activity of the baseline[28]. We found that the average firing rates during the introduction ($7.48 \pm 0.42$ spike/s), chase ($9.11 \pm 0.44$ spike/s), and attack ($7.99 \pm 0.34$ spike/s) phase were significantly higher than the baseline ($5.67 \pm 0.27$ spike/s) (Fig. 1d). Even during failed chases and attacks, the average firing rates were significantly higher than the baseline (Supplementary Fig. 2d, e). In comparison, the firing rates during the eating phase were much lower than the baseline ($3.84 \pm 0.20$ spike/s) (Fig. 1d). The proportions of excited and inhibited neurons in the introduction phase were similar (282/618 excited, 243/618 inhibited) (Fig. 1e, f and Supplementary Fig. 2f). In contrast, most neurons were excited in the chase phase (407/618 excited, 136/618 inhibited) and attack phase (390/618 excited, 130/618 inhibited) (Fig. 1e, f and Supplementary Fig. 2f). Most neurons were inhibited in the eating phase (97/618 excited, 402/618 inhibited) (Fig. 1e, f and Supplementary Fig. 2f). Also, we found that the change in firing rates of individual neurons compared to baseline firing during the introduction, chase, or attack phase was not correlated with the baseline, while the change of firing rates during the eating phase had a weak negative correlation with the baseline (Supplementary Fig. 5i–l). Then, we performed the analyses of velocity curves during each phase and found a subset of LPAG neurons activated during different hunting phases were also sensitive to the movement (Fig. 1f and Supplementary Fig. 2g). Correlation analysis revealed that the activity of these movement-activated neurons during the prey chase, but not during the prey introduction, attack and eating phases, was linearly correlated with the movement speed (Supplementary Fig. 2g).

Next, to explore the relationship between the neuronal activity dynamics and the individual behavioral phase, we calculated the normalized activity of each phase across neurons and categorized neurons based on a hierarchical clustering algorithm[29–31]. Interestingly, the recorded neurons in the LPAG could be separated into seven distinct clusters (Fig. 1g) and each cluster had a firing preference to specific hunting phases (Fig. 1h). For example, both type I and type IV neurons fired preferentially to the prey introduction. Yet, type I neurons showed a much larger response than type IV neurons and maintained high activity during the chase and attack phases (introduction-related cells, Fig. 1h). In contrast, the activity of type IV neurons diminished after the prey introduction. Thus, we speculate that type I neurons might serve a role in motivation that drives the whole hunting process, while type IV neurons might only function in prey detection. Both type III and type VI neurons fired preferentially to the prey chase. Yet, type III neurons showed a much larger response than type VI neurons and maintained high activity in the attack phase (chase-related cells, Fig. 1h). In contrast, the activity of type VI neurons diminished after the prey chase. Thus, we reason that type III neurons might be recruited for the movement speed and type VI

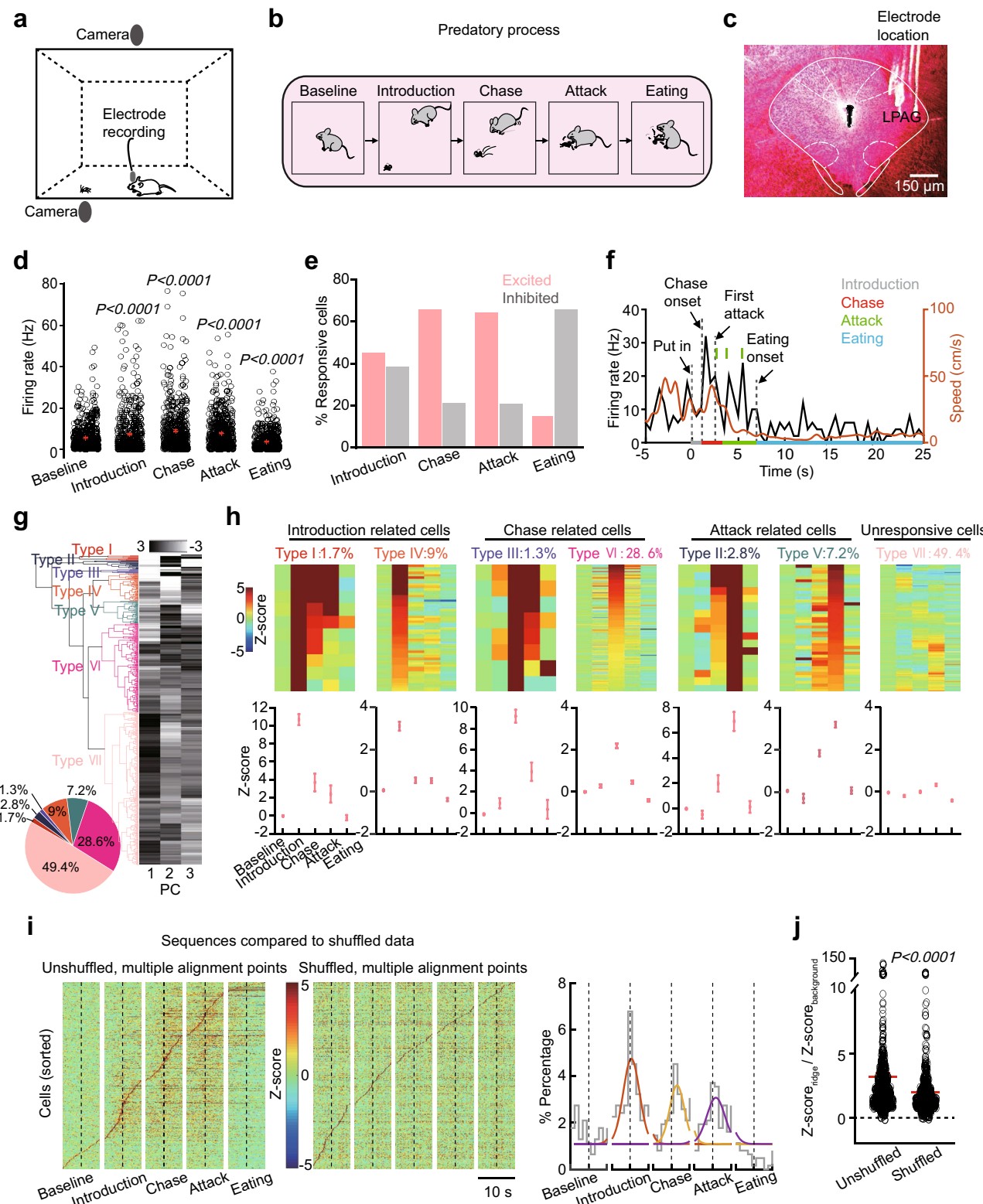

neurons might be recruited selectively for the prey chase. Both type II and type V neurons preferentially fired during the attack phase. Yet, type II neurons showed a much larger response than type V neurons. Also, both types maintained high activity in the chase phase (attack-related cells, Fig. 1h). Thus, we speculate that type II and type V neurons might encode different attack actions, respectively. Type VII neurons did not have a significant response to all phases (Fig. 1h).

To check whether the neuronal firing pattern was time-locked to behavioral phases, we aligned recorded cells according to their peak firing rates. Remarkably, we found that the active phases of each neuron were interlaced in time, forming a sequence chain of activity spanning the entire predatory process (Fig. 1i and Supplementary Fig. 2h). In contrast, there was no similar time sequence in shuffling datasets according to previous analytical methods[32], suggesting that this sequential neuronal firing pattern

**Fig. 1 Distinct clusters of LPAG neurons are sequentially recruited in predatory hunting. a** Scheme of the in vivo single-unit recording system for predatory hunting behavior. **b** Scheme of the hunting process. **c** Histology of electrode trace in the LPAG. Scale bar, 150 μm. **d** Firing rates of all the neurons recorded during different predatory phases ($n = 618$ neurons, two-sided Wilcoxon's signed-rank test). **e** Percentage of responsive cells during different predatory phases. **f** A typical neuronal activity trace and velocity curve during different predatory phases. Each vertical line represents one attack segment. Dotted lines represent the timepoints of put-in of crickets, chase onset, first attack, and eating onset, respectively. **g** Principal component analysis (PCA) and hierarchical clustering dendrogram classified neurons into seven types. **h** Z-score of firing rates in different predatory phases for each of the seven types of neurons (type I: $n = 8$; type II: $n = 17$; type III: $n = 8$; type IV: $n = 55$; type V: $n = 44$; type VI: $n = 177$; type VII: $n = 305$ neurons). The z-score heatmap for each neuron was shown at the top. **i, j** Comparison of sequences to shuffled data. **i** Left, normalized responses of all recorded neurons during predatory hunting, sorted by their peak responses. Middle, shuffled datasets from the same set of neurons as in the left panel (based on 1000 shuffles). Right, the histogram of the percentage of neurons with a response peak in the distribution during different predatory phases. Different colored curves represent peak fitting of the distribution of maximum points across hunting phases. Dotted lines (**i, j**) represent the timepoints of put-in of crickets, chase onset, first attack, and eating onset, respectively. **j** Ridge-to-background Z-score ratio for the plots from **i** ($n = 618$ neurons, two-sided Wilcoxon's signed-rank test). The ridge was defined as the mean z-score in five bins surrounding the peak value, and the background was defined as the mean z-score in all data points. The ridge-to-background ratio provides a measure of how selective the activity of the cell was during hunting. Data are presented as the mean ± SEM. P values are indicated in all panels. Source data are provided as a Source Data File.

during hunting was not an artifact produced by data sorting (Fig. 1i, j). In the activity histogram, there were three peaks in the phases of prey introduction, chase, and attack, which were time-locked to the prey introduction, the chase onset, and the first attack, respectively (Fig. 1i). Consistent with the order of the sequence, principal component analysis (PCA)-clustered cells corresponded to the peak response-aligned cell groups (Supplementary Fig. 2i). Together, these results indicate that LPAG neuron ensembles display a sequential activity pattern to the prey target, and the activity is time-locked to different hunting actions.

**LPAG neurons are recruited for sensory detection, risk assessment, and target discernment.** To discover more specific behavioral actions of mice displayed during the prey introduction, we performed single-unit recordings in head-fixed mice and delivered visual and auditory stimuli which occurred during the cricket introduction (Supplementary Fig. 3a). We found that a subset of LPAG neurons was excited by the moving visual stimulus, while few recorded neurons were sensitive to the static visual spot or the auditory stimulus (Supplementary Fig. 3b, c), suggesting that LPAG neurons detect prey motion signals during the introduction phase.

Moreover, mice need to assess whether the prey target was dangerous during the introduction phase. Therefore, we ran a risk-assessment assay by presenting an awake rat to the hunting mouse and simultaneously recorded single-units in the LPAG (Supplementary Fig. 3d). Interestingly, we found that 47.1% of assessment cells were introduction cells, while only a small percentage of introduction cells (8.8%) were risk-assessment cells (Supplementary Fig. 3e, f). Therefore, these results demonstrate that introduction cells are recruited for risk assessment.

To check whether mice would discern targets via the LPAG, we recorded 148 neurons during interaction with high fat-food diet (HFD) pellets and small textured objects (Supplementary Fig. 4a, b) and found that mice ran significantly faster toward crickets than that to HFD and objects (Supplementary Fig. 4c). A similar percentage of LPAG neurons (~50%) was activated by introducing either crickets, HFD, or objects, while slightly fewer neurons were inhibited by the cricket introduction (Supplementary Fig. 4d). In the chase phase, slightly more neurons were excited and fewer neurons were inhibited in response to crickets as compared to HFD and objects (Supplementary Fig. 4e). During eating, the majority of LPAG neurons (80%) were inhibited during the eating of crickets and HFD (Supplementary Fig. 4f). The proportion of excited and inhibited neurons by object sniffing was about the same (Supplementary Fig. 4g). These data indicated that LPAG neurons discerned incentive targets over non-prey objects. Nevertheless, we analyzed the neuronal firing

sequence pattern to HFD and objects. We found a weaker sequential firing pattern to HFD than prey and found no sequential firing pattern to objects (Supplementary Fig. 4h–j), suggesting that the sequential activity pattern is unique to the prey target. Therefore, these results demonstrate that the introduction phase is a complicated behavioral process where LPAG neurons were recruited for sensory detection, risk assessment, and target discernment.

**LPAG^Vgat and LPAG^Vglut2 neurons are differentially recruited in different hunting phases.** To identify the genetic signature of neuron ensembles activated in each behavioral phase, we recorded LPAG^Vgat and LPAG^Vglut2 neurons by optrode (Fig. 2a). First, we tagged the two major cell types with ChR2 by crossing Vgat-IRES-Cre or Vglut2-IRES-Cre with ChR2 reporter (Ai-32) mice or by injecting Cre-inducible adeno-associated virus (AAV) expressing ChR2 into the LPAG of the Vgat-IRES-Cre or Vglut2-IRES-Cre mice (Fig. 2a and Supplementary Fig. 1b). Then, we delivered 5-ms laser pulses of blue light (473 nm, 8–10 mW, 20 Hz) to premise short-latency action potentials in ChR2-expressing neurons (Fig. 2b) and identified a light-responsive unit as follows: laser-associated spike latency test (SALT) of P values ($P < 0.01$ and Supplementary Fig. 5a), the high similarity between laser-evoked and spontaneous spikes (correlation coefficient >0.85; Supplementary Fig. 5b), high reliability (>0.6; Supplementary Fig. 5c), short latency and low jitter (Supplementary Fig. 5d)[33–35]. A representative trace for each genotype was shown in Fig. 2c, d, respectively. Noticeably, the baseline and peak firing rates of LPAG^Vgat neurons were significantly higher than those of LPAG^Vglut2 neurons (Supplementary Fig. 5e, f). We found that the LPAG^Vgat and LPAG^Vglut2 neurons were not separable from each other by using half-width and trough-to-peak as reported before[36] (Supplementary Fig. 5g, h).

Next, to further categorize LPAG^Vgat and LPAG^Vglut2 neuronal responses in each hunting phase, we performed the same hierarchical clustering as in Fig. 1g. All tagged neurons fell into distinct clusters according to response profiles (Fig. 2e–h). As expected, most of the recorded cells belonged to nonresponsive type VII cells. Notably, the responsive LPAG^Vgat neurons were primarily activated during the prey chase (31.4%, type VI), and the rest were type I cells (8.6%) that maintained high activity in the prey introduction, chase, and attack phases (Fig. 2f). In contrast, the responsive LPAG^Vglut2 neurons were primarily activated during the attack phase (type II and V; Fig. 2h). Neither type III nor type IV clusters were detected, which was probably due to low representation and limited total recorded cell numbers. Overall, our data suggest that neurons recruited at different hunting phases are genetically separable. LPAG^Vgat

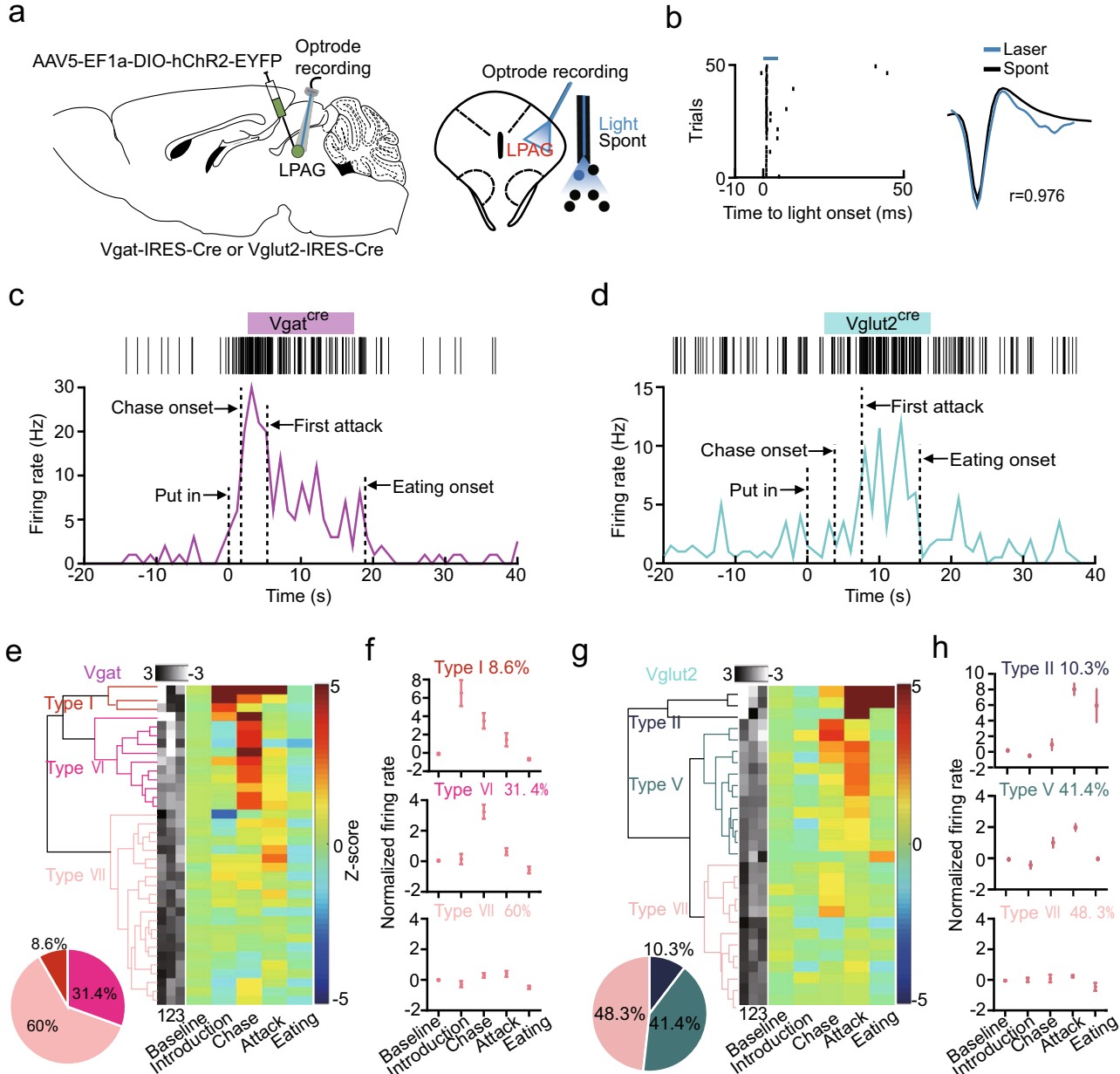

**Fig. 2 LPAG^Vgat and LPAG^Vglut2 neurons are differentially recruited in different hunting phases. a** Scheme of in vivo optrode recordings. **b** Left, spike raster for laser-evoked spikes of a ChR2-tagged neuron. The light pulse was shown in blue (5-ms duration). Right, comparison between light-evoked (blue) and spontaneous (black) spike waveform. **c** An example of the optogenetically identified LPAG^Vgat neuron that fired during different predatory phases. **d** An example of the optogenetically identified LPAG^Vglut2 neuron that fired during different predatory phases. **e** PCA and hierarchical clustering dendrogram classified LPAG^Vgat neurons into three types. The z-score activity plots of 35 LPAG^Vgat neurons tagged during different predatory phases were shown. **f** Z-score of firing rates in different predatory phases for each of the three types of LPAG^Vgat neurons (type I: $n = 3$; type VI: $n = 11$; type VII: $n = 21$ neurons). **g** PCA and hierarchical clustering dendrogram classified LPAG^Vglut2 neurons into three clusters. The z-score activity plots of 29 LPAG^Vglut2 neurons tagged during different predatory phases were shown. **h** Z-score of firing rates in different predatory phases for each of the three types of LPAG^Vglut2 neurons (type II: $n = 3$; type IV: $n = 12$; type VII: $n = 14$ neurons).

neurons are recruited in the introduction, chase, and attack phases, while LPAG^Vglut2 neurons are mainly recruited in the attack phase.

**LPAG^Vgat neurons are necessary for both prey chase and attack.** The observation that LPAG^Vgat neurons were recruited during hunting phases suggests that these neurons might be functionally important. To test this, we bilaterally injected Cre-inducible archaerhodopsin (ArchT), a light-sensitive inhibitory proton pump (AAV5-EF1a-Flex-ArchT-GFP), into the LPAG of

Vgat-IRES-Cre mice and implanted optic fibers bilaterally into the LPAG (Fig. 3a and Supplementary Fig. 1c). We found that LPAG^Vgat neurons infected by ArchT were about 39.6% of total LPAG cells (Supplementary Fig. 6a, b) and about 88.4% of Vgat+ neurons by ArchT infection were localized to the LPAG (Supplementary Fig. 6f). Viruses expressing EYFP only were used as controls. Mice were trained to adapt to crickets before phase-specific photoinhibition (Fig. 3b). Strikingly, photoinhibition of LPAG^Vgat neurons during the chase phase substantially prolonged the latency to attack and increased the number of chase

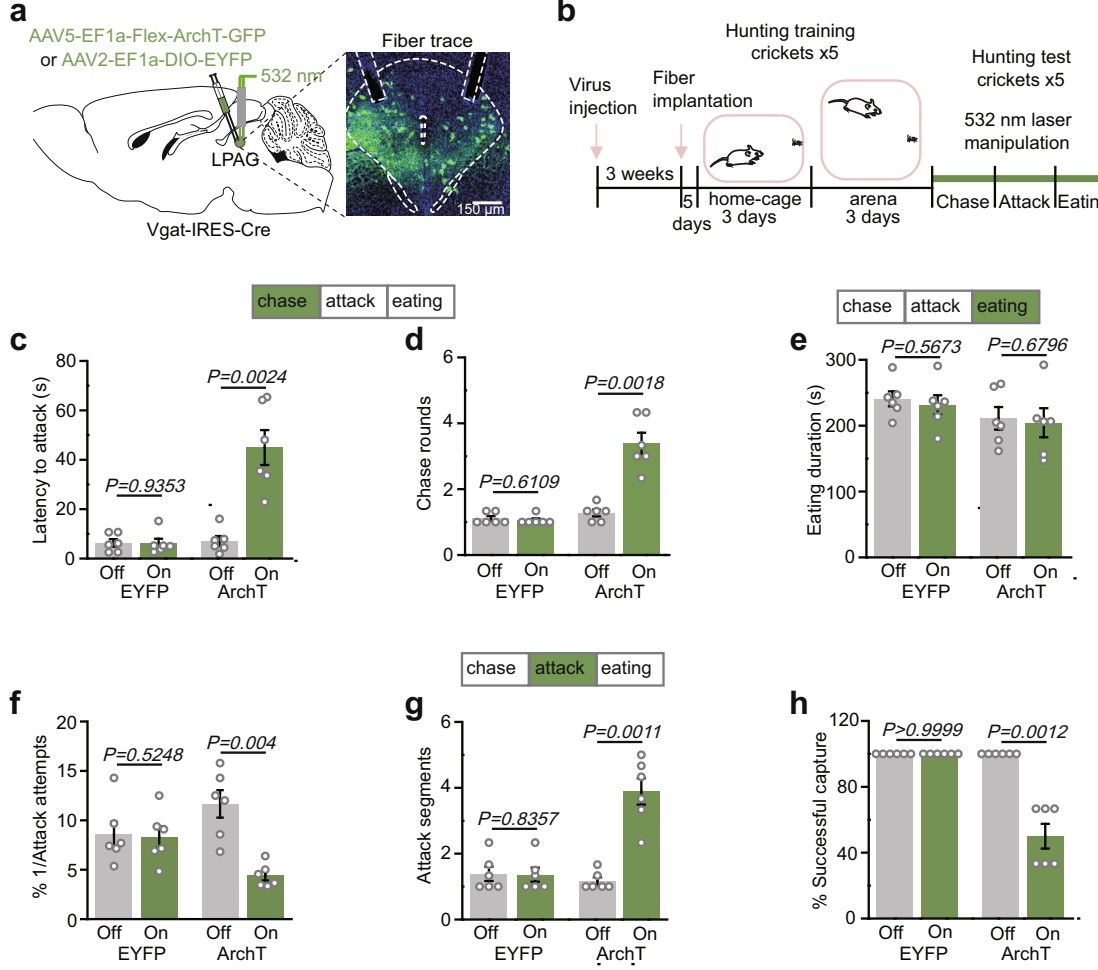

**Fig. 3 Phase-specific inhibition of LPAG[Vgat] neurons suppresses hunting. a** Left, scheme for photoinhibition of LPAG[Vgat] neurons. Right, histology of injection sites and fiber traces in the LPAG ($n = 6$ mice). Scale bar, 150 μm. **b** Experimental protocol of the photoinhibition. **c, d** Changes in latency to attack (**c**) and chase rounds (**d**) after bilateral photoinhibition of LPAG[Vgat] neurons in the chase phase ($n = 6$ mice for each group, two-tailed paired $t$ test). The laser (532 nm, 12 mW, continued) was turned on between the chase onset and the first attack and turned off if mice left the prey. Latency to attack was the time taken from the cricket introduction to the first attack. Chase rounds were the number of prey chases before the first attack. **e** Changes in eating duration after bilateral photoinhibition of LPAG[Vgat] neurons in the eating phase ($n = 6$ mice for each group, two-tailed paired $t$ test). The 532-nm laser (12 mW, 30-s on/30-s off cycles) was turned on when the mouse began to eat the cricket and turned off when the mouse stopped eating. The eating duration was the time taken from the beginning of eating until the eating stopped. **f–h** Changes in attacking efficiency (**f**), attack segments (**g**), successful rate (**h**) after bilateral photoinhibition of LPAG[Vgat] neurons in the attack phase ($n = 6$ mice for each group, two-tailed paired $t$ test). 532-nm laser (12 mW, continued) was turned on between the first attack and successful prey capture or failure (6 min time out). Attack efficiency was 1/attack attempts before capture, and attack attempts were the sum of jaw and forepaw attacks during the whole attack phase. Each segment ends when mice leave the prey. The capture success rate was the percentage that the mouse successfully captured crickets. Data are presented as the mean ± SEM. $P$ values are indicated in all panels. Source data are provided as a Source Data File.

trials per capture (Fig. 3c, d), which is consistent with their neuronal activity (Fig. 2f). In contrast, there was no significant change in eating time after photoinhibition (Fig. 3e). Interestingly, inhibition of LPAG[Vgat] neurons during the attack phase also significantly decreased predatory efficiency by reducing the attack efficiency (1/number of attacks per capture; Fig. 3f), increasing attack segments per capture (each segment ends when mice leave the prey; Fig. 3g), and reducing the capture success rate (Fig. 3h). We did not calculate the capture duration after photoinhibition since most of the capture was not successful within the 6-min time limit (Fig. 3g). Photoinhibition via ArchT had no significant effect on the chase speed and spontaneous movement of LPAG[Vgat] mice (Supplementary Fig. 6c, g–i). Thus, these data indicate that LPAG[Vgat] neurons are required for both chase and attack, which is consistent with the increased neuronal activity during the chase and attack phase (Fig. 2f).

The observation that LPAG[Vgat] neurons are required for both chase and attack led us to suspect that these neurons are important for inducing positive motivation that supports hunting[16]. Therefore, we performed the real-time place preference (RTPP) assay, where mice in one side of the test chamber received a laser stimulation to photoactivate LPAG[Vgat] neurons (Supplementary Figs. 1d, 7a, and 8a–d). As expected, LPAG[Vgat-ChR2] mice showed a significant preference to the laser stimulation side (Supplementary Fig. 7b–d), suggesting a positive valence after photoactivation of these neurons. Next, we performed an intracranial optical self-stimulation (ICSS) test where mice performed nose poke to receive laser stimulation (Supplementary Fig. 7e). Notably, LPAG[Vgat-ChR2] mice exhibited significantly more nose poking to obtain the laser reward and significantly higher breakpoint than LPAG[Vgat-EYFP] mice (Supplementary Fig. 7f–j). Then, we explored whether LPAG[Vgat] neurons were

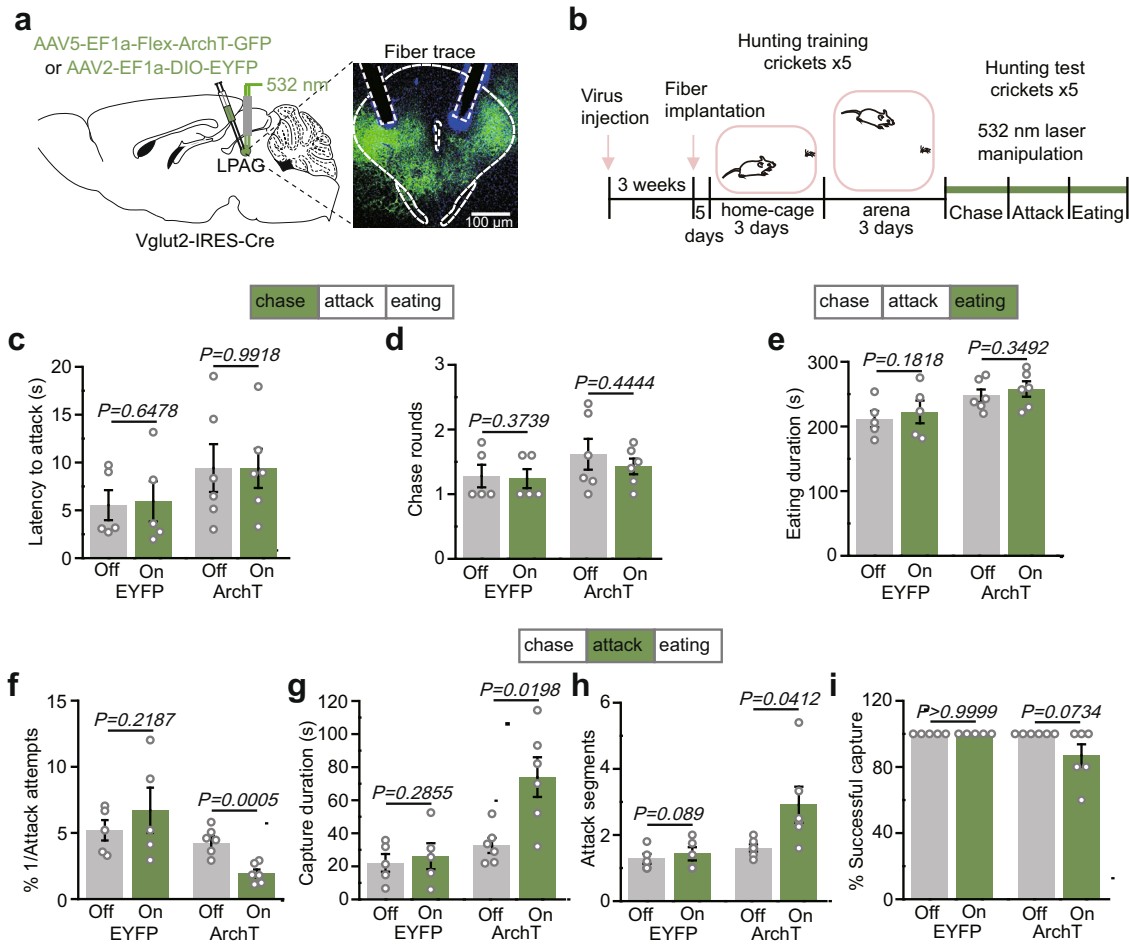

**Fig. 4 Phase-specific inhibition of LPAG Vglut2 neurons suppresses predatory attack. a** Left, scheme for photoinhibition of LPAG Vglut2 neurons. Right, histology of injection sites and fiber traces in the LPAG ($n = 6$ mice). Scale bar, 100 μm. **b** Experimental protocol of the photoinhibition. **c, d** Changes in latency to attack (**c**) and chase rounds (**d**) after bilateral photoinhibition LPAG Vglut2 neurons in the chase phase ($n = 6$ mice for ChR2 group, $n = 5$ mice for EYFP group, two-tailed paired t test). The laser (532 nm, 12 mW, continued) was turned on during the chase phase. Latency to attack was the time taken from the cricket introduction to the first attack. Chase rounds were the number of prey chases before the first attack. **e** Changes in eating duration after bilateral photoinhibition of LPAG Vglut2 neurons in the eating phase ($n = 6$ mice for ChR2 group, $n = 5$ mice for EYFP group, two-tailed paired t test). The 532-nm laser (12 mW, 30-s on/30-s off cycles) was turned on during the cricket eating phase. The eating duration was the time taken from the beginning of eating until the eating stopped. **f–i** Changes in attacking efficiency (**f**), capture duration (**g**), attack segments (**h**), and successful rate (**i**) after bilateral photoinhibition of LPAG Vglut2 neurons in the attack phase ($n = 6$ mice for ChR2 group, $n = 5$ mice for EYFP group, two-tailed paired t test). The 532-nm laser (12 mW, continued) was turned on during the attack phase. Attack efficiency was 1/attack attempts before capture, and attack attempts were the sum of jaw and forepaw attacks during the whole attack phase. Capture duration was the time from the first attack until the mouse successfully captured the cricket. Each segment ends when mice leave the prey. The capture success rate was the percentage that the mouse successfully captured crickets. Data are presented as the mean ± SEM. P values are indicated in all panels. Source data are provided as a Source Data File.

necessary for free-reward (10% sucrose) consumption (Supplementary Fig. 7k). Indeed, photoinhibition of LPAG Vgat neurons nearly abolished licking (Supplementary Fig. 7l, m). Altogether, these data indicate that LPAG Vgat neurons are sufficient to induce a positive motivational value that might be essential for hunting.

**LPAG Vglut2 neurons are necessary for the predatory attack.** To study the function of LPAG Vglut2 neurons in hunting, we bilaterally injected AAV5-EF1a-Flex-ArchT-GFP into the LPAG of Vglut2-IRES-Cre mice and implanted optic fibers bilaterally into the LPAG (Fig. 4a and Supplementary Fig. 1c). We found that LPAG Vglut2 neurons infected by ArchT were about 35.3% of total LPAG cells (Supplementary Fig. 6d, e), and about 89.5% of Vglut2+ neurons by ArchT infection were localized to the LPAG (Supplementary Fig. 6f). Mice were trained to adapt to crickets before tests (Fig. 4b). Photoinhibition of LPAG Vglut2 neurons during the chase phase did not significantly affect behavioral

parameters, including latency to attack and chase rounds (Fig. 4c, d). Meanwhile, there was no significant change in eating time (Fig. 4e). However, inhibition of LPAG Vglut2 neurons during the attack phase significantly impaired hunting efficiency, including reducing attack efficiency and increasing capture duration (Fig. 4f–h). Interestingly, it did not affect the capture success rate (Fig. 4i). Therefore, these data indicate that LPAG Vglut2 neurons are selectively required for an attack, which is quite consistent with their neuronal activity pattern (Fig. 2h). We did not test whether photoactivation of LPAG Vglut2 neurons would affect hunting as photoactivation of these neurons induced strong defensive behaviors as reported before[18], which would mask the hunting behavior (Supplementary Figs. 1d and 9a–e).

As LPAG Vglut2 neurons are selectively required for an attack, we wondered whether their neuronal activity is time-locked with masseter muscle activity. Thus, we simultaneously recorded the activity of LPAG Vglut2 neurons ($n = 22$ units) and masseter

muscle electromyography (EMG) activity when mice were attacking crickets (Supplementary Fig. 10a). To identify whether attack-activated LPAG$^{Vglut2}$ neurons correlated to the masseter activity, we reviewed the relationship between spikes of LPAG$^{Vglut2}$ neurons and the EMG activity as the previous study[37]. We calculated the average value of the EMG signals within the 800-ms window around each spike of LPAG$^{Vglut2}$ neurons recorded during the attack phase. Notably, we found a substantial number (6/22) of LPAG$^{Vglut2}$ neurons whose activities were time-locked with masseter EMG, further validating the function of these neurons in attack (Supplementary Fig. 10b–e). Moreover, LPAG$^{Vglut2}$ neurons might not be important for encoding motivation as its photoinhibition did not affect free-reward consumption (Supplementary Fig. 7m).

**LPAG afferents differentially regulate the neuronal activity in different hunting phases.** The LPAG receives GABAergic inputs from key brain regions involved in hunting, including the CeA, the LH, and the ZI. To assess functional connectivity of CeA/LH/ZI-LPAG pathway, we unilaterally injected AAV-hSyn-FLEX-EGFP-WPRE-hGHpA into the LPAG of Vgat-IRES-Cre mice and injected AAV-hEF1a-DIO-hChR2 (H134R)-mCherry into the CeA, the LH and the ZI, and recorded postsynaptic currents (PSCs) from Vgat+ and Vgat− (presumably glutamatergic) neurons in the LPAG by optogenetically activating CeA/LH/ZI-LPAG projections, respectively (Supplementary Fig. 11a, d, g). We found that both Vgat+ and Vgat− neurons recorded in the LPAG received monosynaptic inhibitory inputs from the CeA, the LH, and the ZI (Supplementary Fig. 11b, c, e, f, h–j). To anatomically quantify LPAG-projecting CeA/LH/ZI neurons, we used both Flp- and Cre-dependent fluorescent proteins to quantify the overlapping between any two types out of the three projecting neurons (Supplementary Fig. 12a–c). Interestingly, we found that only a small percentage of overlap between any of the two types of projecting neurons, including the CeA and the LH (Supplementary Fig. 12d, g), the CeA and the ZI (Supplementary Fig. 12e, h), and the LH and the ZI (Supplementary Fig. 12f, i). Also, we quantified the cell numbers at different bregma sites of the LPAG (Supplementary Fig. 12j) and did not notice an apparent difference in bregma sites. Altogether, these data indicate that the CeA, the LH, and the ZI are functionally and anatomically connected with distinct subpopulations of LPAG neurons, respectively.

How these GABAergic inputs affect the formation of the dynamic predatory sequence in the LPAG is largely unknown. To test the effect of distinct inputs on the hunting sequence, we performed in vivo single-unit recordings on LPAG neurons after input-specific lesions during predatory hunting (Fig. 5a–c). To achieve input-specific lesions, we bilaterally injected AAV-retro-DIO-Flippase into the LPAG of Vgat-IRES-Cre mice and injected AAV2/9-hEF1a-fDIO-taCaspase3-TEVp into the CeA, the LH, and the ZI, respectively. The flippase injected from the LPAG will travel retrogradely back to the soma to drive the expression of taCasp3, thus killing neurons in these three input areas, respectively (Fig. 5a–c and Supplementary Fig. 13a–g). After lesions and recordings, we sorted the neurons into different groups following the same criteria as Fig. 1g (Fig. 5d–f). Interestingly, we found that the ablation of LPAG-projecting GABAergic neurons from the CeA reduced the ratio of type I introduction-related cells, while increasing the ratio of type IV introduction-related cells (Fig. 5g). Both the ratios of both type VI chase-related cells and type V attack-related cells were nearly abolished, while the ratio of type II attack-related cells was increased (Fig. 5g). Moreover, the overall firing rates of LPAG neurons were significantly reduced (Supplementary Fig. 13h). These data demonstrate that ablation of LPAG-projecting

GABAergic neurons from the CeA mainly reduces the activity during the chase and attack phases and affects the activity in the introduction phase.

For the ablation of LPAG-projecting GABAergic neurons from the LH, it substantially reduced the ratios of both attack-related neurons (type II and type V) besides slightly increasing the ratio of type IV introduction-related cells (Fig. 5h). It did not significantly affect other types of cells, suggesting that removing the LH GABAergic inputs selectively affects attack. Meanwhile, overall firing rates of LPAG neurons were significantly increased (Supplementary Fig. 13i).

For the ablation of LPAG-projecting GABAergic neurons from the ZI, it significantly reduced the ratio of type I introduction-related cells, type VI chase-related cells, and type II attack-related cells, leaving other hunting-related types unchanged (Fig. 5i). Also, it increased the unresponsive cell ratio (Fig. 5i). The firing rates of LPAG neurons were significantly decreased (Supplementary Fig. 13j). These data indicate that LPAG-projecting GABAergic neurons from the ZI mainly regulate the neuronal activity in the introduction and chase phases, and also affect the neuronal activity in the attack phase. Thus, distinct GABAergic inputs onto LPAG neurons differentially affect neuronal activity in predatory phases.

To examine whether the sequential activity pattern of LPAG neurons was affected by input-specific ablations, we ordered the activity of all recorded neurons according to their firing peaks. The *wt* control was replotted from Fig. 1i. Manifestly, after ablation of the CeA inputs, only the peak in the introduction phase was left (Fig. 5k and Supplementary Fig. 13k, l). After ablation of the LH inputs, only the small peak in the chase phase was left (Fig. 5k and Supplementary Fig. 13k, l). After ablation of the ZI inputs, the peak in the attack phase was disappeared (Fig. 5k and Supplementary Fig. 13k, l). Altogether, these findings show that the LPAG as a hub receives distinct inhibitory inputs to regulate predation. The CeA, the LH and the ZI provide distinct input signals to modulate the neuronal activity in distinct hunting phases, but all of these inputs are essential for maintaining the sequential activity pattern intact.

**Lesions of LPAG afferents impair predatory hunting.** The CeA, the LH, and the ZI provide GABAergic inputs to modulate LPAG neuronal activity during hunting. However, how these inputs affect predation in the LPAG is not clear. Therefore, we killed LPAG-projected GABAergic neurons in an input-specific manner (Fig. 6a and Supplementary Fig. 14a, f, k) and tested their impact on hunting tasks. The locomotion activity was not significantly changed after ablation of LPAG-projecting GABAergic neurons from the CeA or the LH (Supplementary Fig. 14b, g and Supplementary Movies 3–6). In contrast, the locomotion activity was significantly reduced after ablation of LPAG-projecting GABAergic neurons from the ZI (Supplementary Fig. 14l and Supplementary Movies 7, 8). These data are consistent with the role of ZI in regulating motivation and the movement speed we reported before[16]. For all three lesions, there was a substantial reduction in predatory efficiency by prolonging the latency to attack, increasing chase rounds (Supplementary Fig. 14d, i, n), reducing attack efficiency, and increasing capture duration (Supplementary Fig. 14e, j, o). Yet, the eating duration was not changed significantly (Supplementary Fig. 14c, h, m).

To further address how individual inputs influence the distinct predatory phase, we compared different hunting parameters across individual brain region lesions. Basically, we found that there was a more substantial effect on prolonging the latency to attack after LH lesions as compared to CeA/ZI lesions (Fig. 6b). The parameters representing the chase and attack, including the

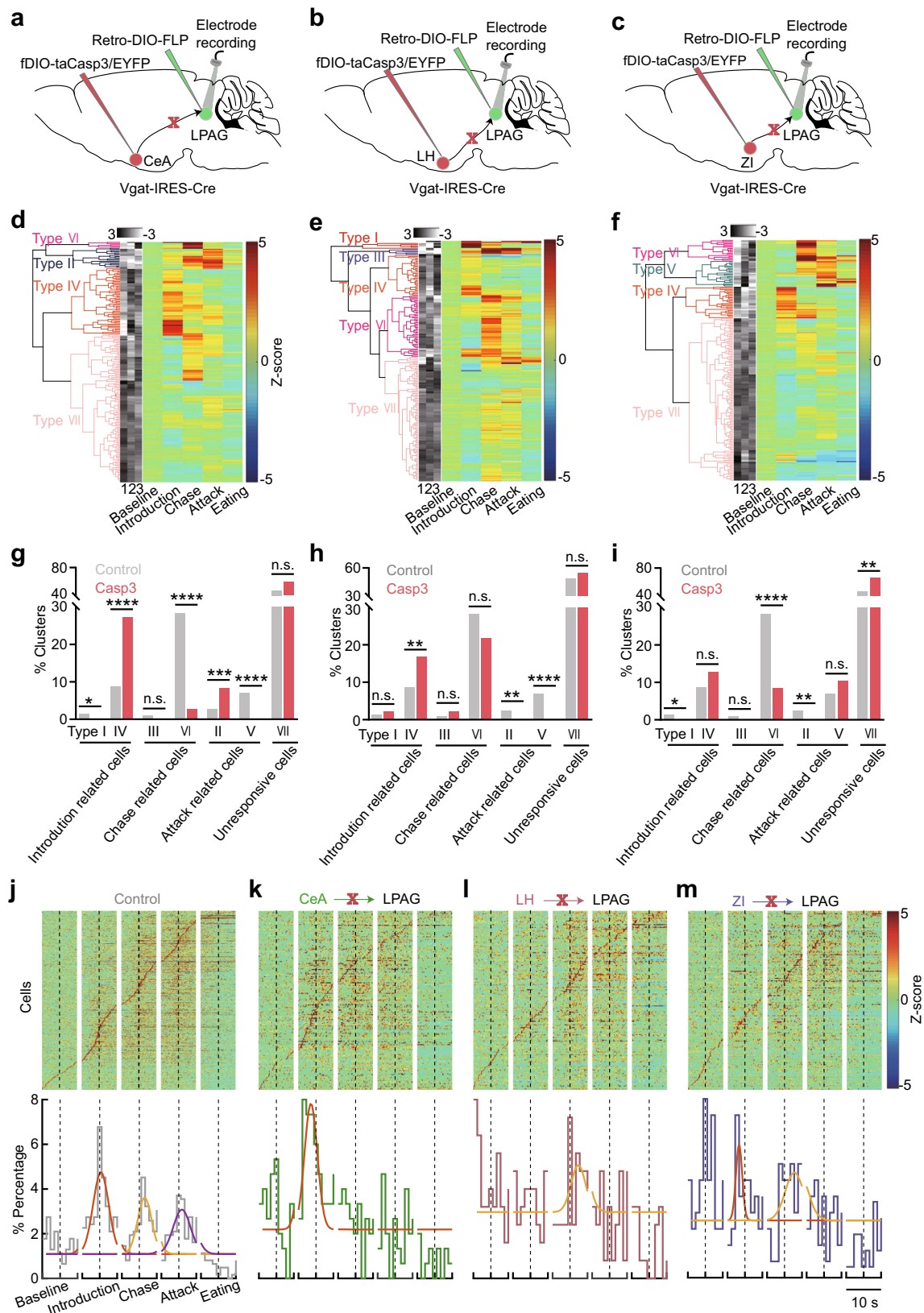

chase numbers, attack segments, and capture duration after CeA lesions, were more severely impaired than those after LH/ZI lesions (Fig. 6c, e, f). These data were consistent with the role of these LPAG-projecting GABAergic neurons from the CeA in mediating pursuit and the biting behavior during hunting[12]. We did not find a significant difference in the attack efficiency after ablation of LPAG-projecting GABAergic neurons from the CeA/LH/ZI (Fig. 6d). The eating duration was not significantly different between these lesions (Fig. 6g). Together, these data conclude that removing either of the CeA/LH/ZI inputs to the LPAG is sufficient to interrupt all hunting motor actions except eating, including prey detection, chase, attack, and capture.

**Fig. 5 LPAG afferents differentially regulate the neuronal activity in different hunting phases. a–c** Experimental scheme of in vivo single-unit recordings in the LPAG after ablation of GABAergic neurons from the CeA (**a**), the LH (**b**), and the ZI (**c**), respectively. **d–f** Principal component analysis (PCA) and hierarchical clustering dendrogram classified neurons into different types after input-defined lesions to LPAG-projecting GABAergic neurons from the CeA (**d**), the LH (**e**), and the ZI (**f**), respectively. **g–i** The percentage of cell types after ablation of LPAG-projecting GABAergic neurons from the CeA ($n = 203$ neurons, two-sided $\chi^2$ test, $P = 0.0352$, $P < 0.0001$, $P = 0.0582$, $P < 0.0001$, $P = 0.0005$, $P < 0.0001$, $P = 0.0553$) (**g**), the LH ($n = 199$ neurons, two-sided $\chi^2$e test, $P = 0.2116$, $P = 0.0023$, $P = 0.2413$, $P = 0.1659$, $P = 0.0092$, $P < 0.0001$, $P = 0.1868$) (**h**), and the ZI ($n = 206$ neurons, two-sided $\chi^2$ test, $P = 0.0342$, $P = 0.1173$, $P = 0.0515$, $P < 0.0001$, $P = 0.0088$, $P = 0.1356$, $P = 0.008$) (**i**), respectively. **j–m**, Top, normalized responses of all recorded neurons during predatory hunting, sorted by the peak response of each neuron during hunting. Bottom, the histogram of the percentage of neurons with a response peak in the distribution during different predatory phases in the control (**j**; replotted from Fig. 1i), and after ablation of LPAG-projecting GABAergic neurons from the CeA (**k**), the LH (**l**), and the ZI (**m**), respectively. Different colored curves (**j–m**) represent peak fitting of the distribution of maximum point across hunting phases. Dotted lines (**j–m**) represent the timepoints of put-in of crickets, chase onset, first attack, and eating onset, respectively. Source data are provided as a Source Data File.

However, the extent of interruptions after removing these inputs are different. Removing ZI inputs has a more substantial effect on the movement speed; removing LH inputs has a more substantial effect on latency to attack; while removing CeA inputs has a more profound effect on the prey chase and attack. As the three lesions all affected the formation of activity pattern dynamics (Fig. 5), we, therefore, propose that the intact sequential activity pattern is necessary for efficient hunting actions.

## Discussion

We demonstrate that distinct subpopulations of LPAG neurons are sequentially recruited in different hunting phases. Specifically, LPAG$^{Vgat}$ neurons are recruited in the prey detection, chase, and attack phases, while LPAG$^{Vglut2}$ neurons are selectively recruited in the attack phase. Accordingly, phase-specific inhibition of LPAG$^{Vgat}$ neurons suppresses chase and attack without affecting eating, whereas phase-specific inhibition of LPAG$^{Vglut2}$ neurons only impairs attack. Meanwhile, we found that the CeA, the LH, and the ZI provide key GABAergic inputs to regulate LPAG neuronal activity during different hunting phases. Selective ablation of either of these inputs interrupts the sequential activity pattern in the LPAG and impairs whole hunting motor actions except eating. Altogether, these data reveal that the sequential activity pattern in LPAG neurons is critical for efficient predation.

**The LPAG encodes the predatory motor sequence**. Sequential encoding of the behavior is well established for memory or decision-making processes in the hippocampus and motor cortex to support important physiological functions[7–11]. For example, activity sequence or manifolds in the motor cortex is important for motor generation[11] and the neuronal activity sequence in the hippocampus functions in navigational planning[7]. However, whether this encoding mechanism exists for instinctive behaviors and how it affects behaviors remain less understood. In general, hunting includes stereotypical actions, such as the prey search, pursuit, attack, and consumption[2,3], which occurs in response to a "releasing signal," and then runs to completion[5]. However, the whole hunting sequence can be flexible and shaped by environmental cues[4]. This sequence is similar to other innate behaviors like self-grooming, mating behavior, maternal behavior, and aggression, where serial of different actions are organized into a chain-like structure[38]. After examining the endogenous activity of LPAG neurons during predatory hunting, we found that they were sequentially activated and time-locked to the sequential motor actions (Fig. 1i). Moreover, these neurons were separated into seven different clusters with an unsupervised method according to their neuronal dynamics (Fig. 1g). Each cluster displays a strongly biased firing to a particular hunting motor action. Cell type-specific recording reveals that LPAG$^{Vgat}$ neurons were activated during the prey detection, chase, and attack,

whereas LPAG$^{Vglut2}$ neurons were primarily activated in the attack phase (Fig. 2). Consistently, inhibition of LPAG$^{Vgat}$ neurons suppressed both chase and attack phases of the predatory process, while inhibition of LPAG$^{Vglut2}$ neurons selectively suppressed the attack phase (Figs. 3 and 4). These results suggest that the LPAG encodes the predatory motor sequence, and the motor sequence could be genetically defined, which is consistent with the concept of "hard-wiring" in instinctive behaviors[39–41].

**LPAG$^{Vgat}$ neurons induce positive motivation**. Motivation plays a pivotal role in predatory hunting[16,27,42]. We have shown previously that the $SC \rightarrow ZI \rightarrow PAG$ circuit integrates multisensory signals and induces a positive motivation to drive hunting[13,16]. Our data showed that LPAG$^{Vgat}$ neurons were activated during the whole hunting process, including the prey detection, chase, and attack (Fig. 2). Activation of LPAG$^{Vgat}$ neurons was appetitive (Supplementary Fig. 7b). Operant assays suggest that activation of these neurons is associated with a strong motivation (Supplementary Fig. 7). Further, photoinhibition of these neurons, but not LPAG$^{Vglut2}$ neurons, blocked free-reward consumption (Supplementary Fig. 7m). Accordingly, inhibition of these neurons suppressed the prey chase and attack, and reduced the hunting success rate (Fig. 3). Therefore, these results suggest that LPAG$^{Vgat}$ neurons induce a strong positive motivation, which is presumably vital in driving hunting. Based on the neural dynamics, type I cells might serve the role in encoding motivation as their activities peak in the prey introduction, but still maintain at a high level in the chase and attack phases (Fig. 2f). Nevertheless, one pitfall is that we did not directly record neuronal activity during motivation tests. Also, it is still possible that inhibition of LPAG$^{Vglut2}$ neurons encodes motivation since these neurons might get inhibited by LPAG inhibitory neurons locally and target the downstream neurons in the LH[43]. More specific genetic manipulations are needed to resolve this issue as broad activation of LPAG$^{Vglut2}$ neurons causes a strong defensive response (Supplementary Fig. 9).

**Different GABAergic inputs differentially contribute to the dynamics of the LPAG neurons during predatory hunting**. Previous studies reveal that the CeA[14], the LH[2], and the ZI[16] provide GABAergic inputs to the L/VLPAG to regulate hunting. However, it is unclear that how these functional projections regulate cell ensemble sequences of LPAG neurons. Overall, we found that blocking GABAergic inputs from the CeA and the ZI decreased the mean firing rate, whereas blocking inputs from the LH increased the mean firing rates (Supplementary Fig. 13). Specifically, blocking GABAergic inputs from the CeA and the ZI vastly decreased the proportion of type I introduction-related cells and type VI chase-related cells (Fig. 5), but selectively reduced one type of attack-related cells (type V for the CeA, type

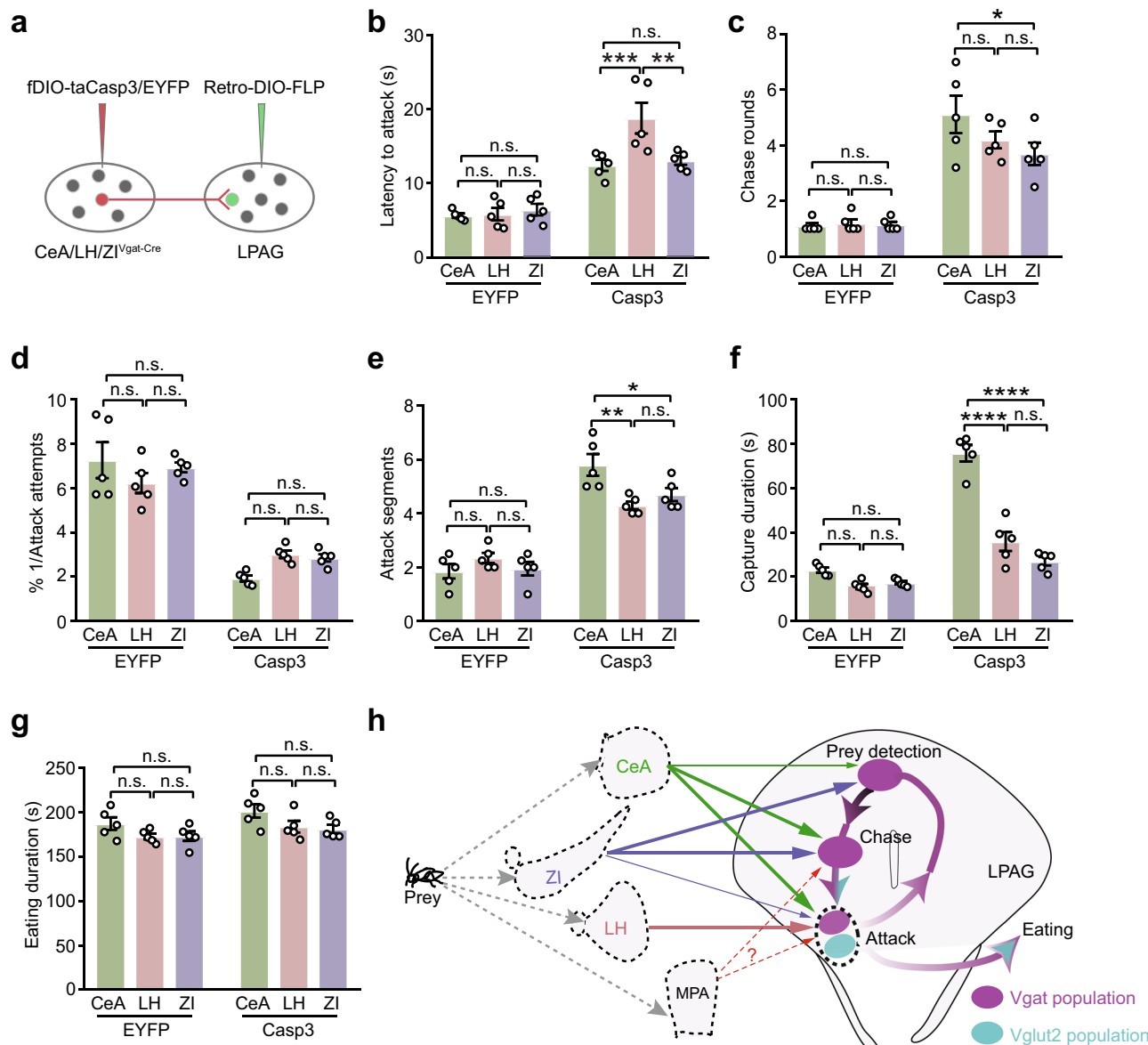

**Fig. 6 Lesions of LPAG afferents impair predatory hunting. a** Experimental scheme of ablation of LPAG-projecting GABAergic neurons from the CeA/LH/ZI. **b** Comparison of the effect on the latency to attack after ablation of LPAG-projecting GABAergic neurons from the CeA/LH/ZI ($n = 30$ mice, two-way ANOVA, Tukey's multiple comparisons test, $P = 0.991$, $P = 0.854$, $P = 0.9123$, $P = 0.0007$, $P = 0.9041$, $P = 0.0021$). **c** Comparison of the effect on the chase rounds after ablation of LPAG-projecting GABAergic neurons from the CeA/LH/ZI ($n = 30$ mice, two-way ANOVA, Tukey's multiple comparisons test, $P > 0.9999$, $P > 0.9999$, $P > 0.9999$, $P = 0.2405$, $P = 0.0284$, $P = 0.9919$). **d** Comparison of the effect on the attack efficiency after ablation of LPAG-projecting GABAergic neurons from the CeA/LH/ZI ($n = 30$ mice, two-way ANOVA, Tukey's multiple comparisons test, $P = 0.1904$, $P = 0.833$, $P = 0.4512$, $P = 0.1576$, $P = 0.2587$, $P = 0.9533$). **e** Comparison of the effect on the attack segments after ablation of LPAG-projecting GABAergic neurons from the CeA/LH/ZI ($n = 30$ mice, two-way ANOVA, Tukey's multiple comparisons test, $P = 0.3874$, $P = 0.9612$, $P = 0.54$, $P = 0.0014$, $P = 0.0185$, $P = 0.54$). **f** Comparison of the effect on the capture duration after ablation of LPAG-projecting GABAergic neurons from the CeA/LH/ZI ($n = 30$ mice, two-way ANOVA, Tukey's multiple comparisons test, $P = 0.117$, $P = 0.2629$, $P = 0.8872$, $P < 0.0001$, $P < 0.0001$, $P = 0.052$). **g** Comparison of the effect on the eating duration after ablation of LPAG-projecting GABAergic neurons from the CeA/LH/ZI ($n = 30$ mice, two-way ANOVA, Tukey's multiple comparisons test, $P = 0.2131$, $P = 0.227$, $P = 0.992$, $P = 0.1128$, $P = 0.0602$, $P = 0.9465$). **h** Summary of functions of LPAG across predatory hunting. A neuronal ensemble sequence is formed in the LPAG across the whole predatory process. Vgat+ population (light purple) in the LPAG encodes the introduction, chase, and attack, while Vglut2+ population (light blue) encodes the attack. The neuronal activity of LPAG clusters across predatory processes depends on GABAergic inputs from the CeA (green line), LH (orange line), and ZI (purple line). GABAergic inputs from the CeA mainly control prey chase and attack, GABAergic inputs from the LH control prey attack, and GABAergic inputs from the ZI mainly control introduction and chase phases. The role of the inputs from the MPA on hunting needs further investigation. Data are presented as the mean ± SEM. Source data are provided as a Source Data File.

II for the ZI). In the activity histogram, blocking CeA GABAergic inputs eliminated both the two peaks in the prey chase and attack, while blocking ZI GABAergic inputs eliminated the peak in the attack phase. Blocking GABAergic inputs from the LH lost attack-related cells (type II and type V), and eliminated both the

two peaks in the prey introduction and attack phases (Fig. 5). These results indicate that the LPAG integrates signals from different upstream brain regions and these inputs differentially regulate the activity dynamics of LPAG neurons. At the behavioral level, we strikingly found that either of these ablations

severely impaired all hunting actions except eating, including the prey detection, chase, and attack (Fig. 6 and Supplementary Fig. 14). Interestingly, comparative analysis revealed that the degrees of impairment were different. Removing ZI GABAergic inputs has a more substantial effect on the movement speed; removing LH GABAergic inputs affects latency to attack more severely while removing CeA GABAergic inputs impairs the prey chase and attack more profoundly. Therefore, different GABAergic inputs differentially regulate the activity dynamics of LPAG neurons and control predatory hunting actions with a bias. We did not directly test the effect after ablation of MPA neurons as it mainly targeted the ventral PAG[17].

**The releasing mechanism for the hunting behavior.** The predatory motor sequence is triggered by a "releasing signal." For example, a moving visual bar parallel to its direction of movement is sufficient to elicit a predatory attack in toads. Interestingly, we have shown that the SC, the ZI, and the PAG are sensitive to moving visual signals (refs. [13,16,44] and Supplementary Fig. 3b) and mice strongly preferred hunting of moving prey/prey-like objects compared to static targets[13]. Further, photoactivation of CeA/LH/ZI → L/VLPAG circuits induced hunting actions to artificial prey and cricket without training[2,14,16], suggesting the whole hunting motor sequence is released after photoactivation. Also, photostimulation of the MPA → vPAG circuit caused mice to vigorously chase moving objects[17], suggesting the prey chase action is released. Therefore, in contrast to amphibians, single sensory signals are unlikely to release (or induce) prey chase or hunting in mice due to the higher cognition during prey recognition[13,17]. However, it will release (or induce) hunting actions after strong artificial photoactivation of the CeA/LH/ZI/MPA → PAG circuits where sensory integration probably occurs (at least for the ZI[16]), suggesting the existence of releasing signals within these circuits.

Here, by ablation of the inputs from the CeA/LH/ZI to the LPAG, we found indeed that blocking either of these inputs severely impaired all hunting actions (Fig. 6 and Supplementary Fig. 14). It is still unclear why blocking either of these different inputs all severely affects hunting actions, and photoactivation of either of these different inputs induces hunting. We propose that LPAG neurons encoding different hunting phases are interconnected and chained in sequence to control hunting (Fig. 6h). Therefore, blocking the activity from one input would interrupt the sequence structure and eventually impair hunting. On the contrary, driving the activity from one input would eventually increase the sequence activity and induce hunting. Consistently, we observed a strong overlapping between neural populations activated by the prey introduction, chase, and attack (Supplementary Fig. 4k). Nevertheless, it is plausible that either of the CeA/LH/ZI areas provides inputs to all hunting phases and the inputs provided by the LH to the prey detection and chase phases might not be detected (Fig. 5g–i). Therefore, strong artificial photoactivation of either of the inputs would drive hunting. Further studies are needed to resolve this issue.

The completion of the predatory sequence requires the continuous presence of prey. As shown in Supplementary Movies 1 and 2, once the prey is removed, the hunting behavior will stop. These data are consistent with the previous study that prey capture sequences in zebrafish are disrupted when we interfere with visual processing using genetic mutants or remove visual cues after the behavior is initiated[4]. Thus, the predatory action sequence is not only stereotypical with the continuous presence of prey but also flexible after altered sensory stimuli.

In summary, the central nervous system provides stereotypical but flexible functional neuronal ensembles to encode instinctive behaviors. When an instinctive behavior generates, a series of neuronal ensembles are activated sequentially, which chains various actions into sequences. The actions are considered to be stereotyped, where one neuronal ensemble activated may induce only one fixed type of action output. In contrast, the behavioral sequence is flexible, where the order of ensembles could be adjusted by earlier experience (motor planning, top-down) and sensory evidence (close-range interaction, bottom-up). In such cases, behavioral sequences may be hierarchically organized and influenced by internal states or sensory stimuli. We reason that in this manner, the behavior is robustly performed with low cost but still retains flexibility. Thus, we provide a framework of neuronal ensemble sequences across predatory hunting, which may shed light on other instinctive behaviors.

## Methods

**Animals**. Experimental subjects were adult male mice (8–16 weeks old). The Vgat-ChR2 mice were obtained from Dr. Josh Huang Lab (Cold Spring Harbor Laboratory, Cold Spring Harbor, NY, USA). The *Vglut2-IRES-Cre*, Vgat-IRES-Cre, and Ai-32 mice (Jackson Laboratories stock numbers: 028863, 028862, and 024109) were purchased from the Jackson Laboratory. C57BL/6J mice were from Beijing HFK Bioscience Co., Ltd. All mice were group-housed and bred in the animal facility at the Wuhan National Laboratory under a constant temperature (22 ± 2 °C), humidity (40–60%), and 12-h light/dark cycle (7:00 a.m. to 19:00 p.m.). All experimental procedures were approved by the Hubei Provincial Animal Care and Use Committee and complied with the experimental guidelines of the Animal Experimentation Ethics Committee of Huazhong University of Science and Technology, ShanghaiTech University, and Shanghai Biomodel Organism Co., China.

**Virus**. AAV5-EF1a-Flex-ArchT-GFP, AAV2-EF1a-DIO-EYFP, and AAV5-EF1a-DIO-ChR2 (H134R)-EYFP were from the University of North Carolina Center Vector Core Facility, USA. AAV-retro-DIO-Flp, AAV2/9-hEF1a-fDIO-taCaspase3-TEVp, AAV2/9-hEF1a-DIO-hChR2 (H134R)-mcherry, AAV2/9-hEF1a-fDIO-EYFP, AAV2/8-hSyn-DIO-mCherry, AAV2/1-hSyn-Flp, AAV2/1-hSyn-Cre, and AAV2/9-hEF1a-fDIO-EYFP-WPRE-PA were purchased from Taitool Bioscience (Shanghai, China). All viruses were stored in aliquots at −80 °C.

**Stereotaxic injections and optical fiber implantation**. Mice were fixed in a stereotaxic injection frame (Narishige Scientific Instrument lab, Japan) after being anesthetized (0.1 ml/10 g) with 2% chloral hydrate and 10% ethylcarbamate in 0.9% NaCl. A measure of 0.25 μl of AAV vectors was delivered with a glass micropipette connected air pressure application (PMI-100 Pressure Micro Injector, Dagan Ltd). Coordinates used for injection were as follows: LPAG, −3.52 mm from bregma, 1.1 mm lateral from midline, and 2.8 mm vertical from the cortical surface with 15° angle; ZI, −1.82 mm from bregma, 1.0 mm lateral from midline, and 4.4 mm vertical from the cortical surface; CeA, −1.22 mm from bregma, 2.25 mm lateral from midline, and 4.75 mm vertical from the cortical surface; LH, −0.9 mm from bregma, 1.0 mm lateral from midline, and 4.8 mm vertical from the cortical surface. After surgery, mice were placed on a heating pad until revival. After the virus injection for 3 weeks, optical fibers (diameter: 62.5 μm) were implanted above the LPAG and secured with dental cement.

**Immunohistochemistry**. Mice were deeply anesthetized and perfused intracardially with phosphate-buffered saline (PBS), followed by 4% paraformaldehyde (PFA, w/v in PBS). Following perfusion, brains were removed and post-fixed in 4% PFA at 4 °C overnight, and then were cut into 40-μm-thick coronal sections using a vibratome (Leica 1200S). To examine the virus expression and fiber locations, brain slices were washed three times with PBS and incubated for 30 min with 4′,6-diamidino-2-phenylindole at room temperature. To identify electrode locations, brain slices were washed three times with PBS and incubated for 30 min with Nissl at room temperature.

**Cricket hunting behaviors**. Details of this assay have also been described in our previous study[16], with minor modifications. Briefly, mice were placed in a transparent acrylic arena (30 cm × 30 cm × 25 cm³) to perform crickets hunting. All crickets hunting behavioral assays were recorded using top and side cameras (Logitech, C1000e, 60 fps). Manual video scoring was performed on a frame-by-frame basis using the Potplayer. Prior to the behavioral test, food-restricted mice (3 g/day) were familiar with crickets in two stages: 3 days of crickets adaptation in the home cage and 3 days of crickets adaptation in the arena with optic fiber, 5 crickets per day. During the experiments, mice were allowed to explore the arena for 10 min freely, and the arena was cleaned before the laser stimulus. For each mouse, cricket hunting assays were repeated for five trials, and one cricket was used in one trial. Each trial began with the introduction of the cricket and ended with

consuming the prey. Then, the arena was cleaned with 75% ethanol to eliminate the cricket residuals and the odor. Crickets, bought from Taobao online store (www.taobao.com), were 2–2.5 cm in size. The predatory process was divided into four phases as follows: introduction (the time point when the cricket was placed on the floor), chase (the mouse oriented and ran toward the prey and speed increased sharply[16] before attacking the cricket), attack (between the first attack and the mouse captured the cricket with the forepaw or the jaw), and eating (the mouse consumed the cricket). The process of interacting with HFD was divided into three phases: introduction (HFD was placed on the floor of the arena), chase (the mouse ran toward HFD), and eating (the mouse consumed the HFD). The process of interacting with an object was divided into three phases: introduction (the object was placed on the floor of the arena), chase (the mouse ran toward the object), and sniff (the mouse smelled the object). All behavioral process was scored using a double-blind method.

For optogenetic inhibition experiments, 532-nm laser (12 mW, continued) was phase manually and phase specifically delivered during the whole chase, capture, and eating phases as indicated. For example, for photoinhibition in the chase phase, 532-nm laser (continued) was turned on when the mouse ran toward the cricket before the attack, and the laser was turned off when the mouse started to attack the cricket with the forepaw or the jaw or left it. For photoinhibition in the attack phase, 532-nm laser (continued) was turned on between the first attack and successful prey capture or failure (6 min time out). For photoinhibition in the eating phase, 532-nm laser (30-s on/30-s off cycles) was turned on when the mouse began to eat the cricket, and the laser was turned off when the mouse stopped eating.

To quantify the efficiency of crickets hunting in mice, we used seven parameters as follows: chase rounds (the number of prey chases before the first attack), latency to attack (the time taken from the introduction of the cricket until the mouse first attacked the cricket with the forepaw or the jaw), attack efficiency (1/attack attempts before capture), "attack segments" (if mice could not capture the prey in a series of continuous attacking attempts [a segment], they may several segments till capture), capture success rate (the percentage that the mouse successfully captured the cricket), capture duration (the time from the first attack until the mouse successfully captured the cricket), and eating duration (the time taken from the mouse beginning to eat the cricket until the mouse stopped eating).

### Head-fixed visual and auditory stimuli.
Head-fixed mice were placed on the spherical treadmill during visual and auditory stimuli.

For the visual stimulus, a 3-cm black dot was generated using a free software Psychophysics Toolbox Version 3 (PTB-3) and displayed on an LCD monitor ($45 \times 25$ cm$^2$, DELL) in front of the mice (25 cm distance). During the test, the soundproof chamber had a dim light emitted from the LCD monitor, and mice were allowed to adapt for 10 min in the chamber with the monitor. The black dot was first presented stationarily for 2 s. Then, it moved from the temporal side to the nasal side for 4 s and moved back from the nasal side to the temporal side for 4 s before disappearing. The side infrared camera (PointGrey, Flea3 firewire, monochrome) was used for video recordings. The visual stimulus was delivered 6–8 times with a 20-s interval for each unit recording.

For the auditory stimulus, mice were given an about 60 dB tone stimulus (0.5 s duration, 5 s interval). Each session contained 30 trials.

### Risk-assessment test.
The risk-assessment test was performed as previously described[19]. Briefly, the test was performed in a two-compartment acrylic arena, which had a large compartment ($30 \times 20 \times 25$ cm$^3$) and a smaller ($30 \times 10 \times 25$ cm$^3$) compartment separated by wire mesh. Mice were placed into the large compartment, while rats were introduced into the smaller one. Each mouse was tested five times with different awake rats. Each risk-assessment event was identified when mice extended their bodies and explored the smaller compartment.

### Open-field test.
To evaluate the locomotor activity, LPAG$^{Vgat-ArchT/EYFP}$ and LPAG$^{Vglut2-ArchT/EYFP}$ mice were placed on an acrylic arena ($30 \times 30 \times 25$ cm$^3$) and tested once in this arena. The test lasted 15 min and included 5 min laser off–5 min laser on–5 min laser off. During the laser period, 532-nm laser stimulation was delivered with 30-s on/30-s off cycles. For CeA/LH/ZI-LPAG$^{Casp3/EYFP}$ mice, the open-field test lasted 5 min. The tracks were recorded with a Logitech C1000e camera and analyzed using idTracker soft (https://www.idtracker.es) offline.

### Defensive behaviors.
The behavioral setup was performed as previously described[18]. Mice were tested on an acrylic arena ($30 \times 30 \times 25$ cm$^3$). For DL/LPAG$^{Vglut2-ChR2/EYFP}$ mice, after 1 min of adaptation, the light was delivered four times for 5 s with a 55-s interval. For LPAG$^{Vglut2-ChR2/EYFP}$ mice, after 1 min of adaptation, the light was delivered for 60 s. Laser pattern: 473 nm, 8–11 mW, 20 Hz, 10-ms pulse. The tracks were recorded with a Logitech C1000e camera and analyzed using idTracker offline.

### Real-time place preference.
The RTPP setup was performed as previously described[45]. The RTPP was administered at the same period (14:00–18:00 p.m.) and performed in a two-same compartment conditioning apparatus ($25 \times 25 \times 50$ cm$^3$). The test contained two sessions over two days. On day 1, mice

were gently placed in the center chamber and allowed to freely explore the entire apparatus for 15 min (pre-test). On day 2, one of the side chambers was defined randomly as the laser-coupled chamber. Laser (473 nm, 20 Hz, 10-ms pulse) was delivered once mice entered the laser-coupled chamber for 15 min (room temperature (RT)). The behavior was recorded from above by a video camera. Videos were analyzed using idTracker software. The preference score was calculated by subtracting the time spent on the laser-coupled side on day 1 (pre-test) from the time spent on the laser-coupled side on day 2 (RT). The preference ratio was calculated by dividing the relative time (in %) the mice spent during RT in the conditioned chamber by the relative time (in %) the mice spent in this chamber during the pre-test (RT/pre-test ratio).

### Intracranial self-stimulation.
The ICSS setup was performed as previously described[46]. The protocol consisted of three sessions over 3 days. Mice were fasted overnight before the experiments. On day 1, LPAG$^{Vgat-ChR2/EYFP}$ mice were placed in an operant conditioning chamber ($25 \times 20 \times 18$ cm$^3$) containing a custom two-port nose-poke system. The active poke was randomly designated, and the other one was the inactive poke. Each nose poke detected was accompanied by illumination of a light-emitting diode (1 s) and a tone (1 s). To encourage exploration, both active and inactive ports were baited 10% sucrose (20 μl/poke). On day 2, mice received the laser stimulation (473 nm, 8–10 mW, 20 Hz, 3-s laser pulse) in the active port on a fixed ratio of 1 for 60 min, whereas the inactive poke had no laser stimulation. On day 3, mice were given a progressive ratio schedule task in which each laser delivery required two additional nose pokes than the previous laser pulse (1, 3, 5, 7, 9, 11…). The total session lasted for 60 min, and breakpoints were defined as the highest number of continuous nose pokes made to complete the last ratio, where a laser pulse was not obtained in 6 min. Nose-poke timestamps were recorded, and sucrose was delivered by the LabState ver1.0 for the operant behavior system.

### Free-reward consumption.
The free-reward consumption setup was performed as previously described[16]. LPAG$^{Vgat-Arch/EYFP}$ and LPAG$^{Vglut2-Arch/EYFP}$ mice were food-restricted to 85–90% of their free-feeding body weight. Mice were trained to freely lick a metal spout for 10% sucrose in daily 30-min sessions until lick count was performed. Stable lick was defined as the number of licks over three continuous days, and varied by <15% compared with the first day. During the tests, mice were given ad libitum food access and tested for 10% sucrose consumption during a 30-min session with constant photoinhibition (532 nm, 12 mW), the laser was turned on 5 s followed by a 25-s break. Lick was recorded by a capacitance-based lick meter (Thinker Tech).

### Slice physiological recordings.
Coronal slices (300 μm thick) were from Vgat-IRES-Cre mice, which were injected with AAV-FLEX-EGFP into the LPAG and with AAV-DIO-hChR2-mcherry into the CeA, the LH, or the ZI, respectively, 4 weeks before. The brain was rapidly dissected and placed in ice-cold oxygenated with 95% $O_2$/5% $CO_2$ cutting solution, containing (in mM): 2.5 KCl, 0.5 CaCl$_2$, 7.2 MgCl$_2$, 25 NaHCO$_3$, 1.1 NaH$_2$PO$_4$, 25 D-glucose, 11 sodium ascorbate, 3 sodium pyruvate, and 97 choline chloride. Brain slices were sectioned using a vibratome (Leica VT1000S, Germany). The slices were incubated at 35 °C in a submerged chamber containing artificial cerebrospinal fluid (ACSF) equilibrated with 95% $O_2$/5% $CO_2$ for 30 min and containing (in mM): 118 NaCl, 2.5 KCl, 2 CaCl$_2$, 2 MgCl$_2$, 26 NaHCO$_3$, 0.9 NaH$_2$PO$_4$, and 11 D-glucose. Recordings were performed with oxygenated ACSF in a recording chamber at a perfusion rate of 1–2 mL/min. Pipettes were filled with intracellular solution containing (in mM): 140 K-gluconate, 0.1 CaCl$_2$, 2 MgCl$_2$, 1 EGTA, 2 ATP K$_2$, 0.1 GTP Na$_3$, and 10 HEPES (pH = 7.25). The resistance of pipettes were 3–6 MΩ. Evoked PSCs from LPAG$^{Vgat}$ neurons (EGFP+) and LPAG$^{Vglut2}$ neurons (EGFP−) were elicited by 0.5 ms blue-light stimulation of axon terminals of CeA/LH/ZI projecting to LPAG infected by ChR2. Excitatory post-synaptic current (EPSCs) and inhibitory postsynaptic current (IPSCs) were recorded when the membrane potential was holding at −70 and 0 mV, respectively. To confirm whether the IPSCs were GABAergic, picrotoxin (50 μM) was added to the ACSF at the end of the recordings.

To test whether the PSCs were monosynaptic currents, tetrodotoxin (1 μM) was added to the ACSF. Then, 4-AP (100 μM) was added to confirm whether the PSCs were recovered. Whole-cell patch-clamp recordings were performed with a MultiClamp 700B amplifier (Molecular Devices, CA, USA) connected to a Digidata 1440A interface (Axon, USA). Data were analyzed using the Clampfit 10.0 software.

### In vivo single-unit recordings and data analysis.
In vivo single-unit recordings and data analyses were performed as described in the previous studies[47,48]. In brief, 16-channel electrodes using 25.4-μm formvar-insulated nichrome wire (cat no. 761500, A-M Systems, USA) were inserted into a screw-driven microdrive. The impedance of the electrode tips was 700–800 kΩ. Neuronal recordings and time-stamps for behavior were performed using the Open Ephys (https://open-ephys.org/) recording system and NeuroLego System (Jiangsu Brain Medical Technology Co., Ltd), band-pass filtered at 300–6000 Hz, and sampled at 30 kHz. During the recordings, the amplitude threshold was set at 50 μV to remove the noise. Electrodes were slowly lowered in a 62.5 μm step for each recording session. To record the activity of LPAG$^{Vgat}$ and LPAG$^{Vglut2}$ neurons, we used a

screw-driven optrode consisting of an optical fiber (62.5 μm diameter) surrounded by 16 microwire electrodes. The end of each electrode was 200–400 μm from the end of the optical fiber. Trains of 50 laser pulses (473 nm, 8–10 mW, 20 Hz, 5-ms pulse duration) were delivered for each behavioral session. A light-responsive neuron was identified if spikes were evoked by laser pulses at high reliability (>0.6), stimulus-associated spike latency test of $P$ values ($P < 0.01$) and if the waveforms of the laser-evoked and spontaneous spikes were highly similar (correlation coefficient >0.85)[33,34].

All data analyses were performed using MATLAB 2014b (The Mathworks, Inc., Natick, MA, USA). Manually spikes sorting was based on the basis of three PCs of waveform and waveform energy features using MClust-4.4. Isolation distance (>20) and L-ratio (<0.1) were calculated to identify neurons[49,50]. A unit that inter-spike intervals spikes count in 2 ms were <1% was contained in the analysis. Moreover, to avoid repeat units in a session, a cross-correlation comparison was performed.

For coinstantaneous hunting recordings and behaviors, the neuronal activity was aligned to crickets introduction, chase onset, the first attack, and eating onset as classified by the offline video analysis. We used double-blind observation for time-stamping specific behavioral phases to neuronal firing patterns. To identify neuronal responses in these phases, the mean firing rate in each phase was compared to the baseline firing distribution (60 s before introduction, according to bootstrap test)[51]. A unit was considered to be excited if its mean firing rate was higher than 95% of the distribution. A unit was considered to be inhibited if its mean firing rate was lower than 95% of the distribution. For HFD eating and objects interaction experiments, analyses were similarly processed. The neuronal activity was aligned to HFD or objects introduction, HFD eating, and object sniffing. To further classify neurons, the z-score was calculated in each phase where the mean firing rate was subtracted by the mean firing rate of baseline and then was divided by the standard deviation of baseline. To identify their significant response to hunting behaviors, PCA and cluster analysis were used to perform to analyze LPAG neurons. In brief, PCA was calculated using the singular value decomposition of the z-score (five dimensions, baseline, introduction, chase, attack and eating). Hierarchical clustering was performed using the first three PCs by a Euclidean distance metric and a complete agglomeration method[29,31].

Data were verified whether the sequence was an artifact of ordering data by compared to shuffled datasets, which has been described in the previous study. Each cell z-score was randomly rearranged to create shuffled datasets. The ridge-to-background ratio was calculated from each cell's z-score. The ridge was defined as the mean z-score in five bins surrounding the peak value, and the background was defined as the mean z-score in all data points.

For visual and auditory stimuli recordings, the neuronal activity was aligned to the beginning of each stimulus. The response periods of different stimuli were as follows: moving black disk (2–10 s) and tone (0–0.5 s). Responsive neurons were identified based on z-score (absolute value >2.5) of any bin during different stimuli.

For risk-assessment recordings, the neuronal activity was aligned to the beginning of assessment onset. Assessment cells were defined that the firing rates during assessment significantly increased as compared to the firing rates at 2 s time window randomly chosen for 20 replicates during the whole recording session (Wilcoxon's rank-sum test, $P < 0.001$).

**Electromyogram electrode, recording, and analysis**. The two formvar-insulated tin-plated copper wires (25 mm diameter) were soldered to a two-pin connector. The tips of the EMG electrode were bared and bent. For implants, the skin of the cheek was cut open to fully expose the masseter. EMG electrodes were then inserted into bilateral masseter muscles and fixed with knots. The skin was sutured, and EMG electrodes were anchored to the skull with dental cement. EMG electrodes were connected to recording headstages via flexible recording cables. Signals were recorded using a Microelectrode AC Amplifier Model 1800 (A-M Systems, USA), filtered (10–500 Hz), and digitized at 1000 Hz by Spike Hound software (http://spikehound.sourceforge.net/). Laser timestamps were collected and recorded via Arduino microcontrollers. EMG signals from the masseter and optrode recordings were recorded simultaneously.

EMG analysis was performed as previously described. Spike-triggered EMG (STEMGs) were calculated based on filtered EMG signals. The calculation method is the average value of the EMG signals in the 800 ms window around each spike of LPAG$^{Vglut2}$ neurons recorded during the attack phase. STEMGs were calculated in 10-s increment during the attack, and downsampled with a 5-ms moving average, and then normalized by the number of spikes. Baseline correction of STEMGs is performed by subtracting the boxcar-filtered version of STEMG for 100 ms from the baseline. Strict criteria for assessing whether STEMGs possessed significant peaks: STEMGs must have at least five continuous points that exceed the 98% confidence interval in 60 ms.

**Statistical analysis**. All statistical analysis was performed using MATLAB 2014b and GraphPad Prism 7. The order of experimental conditions was randomly assigned. A double-blind analysis was applied to behavioral data analysis. The $P$ values were calculated by Wilcoxon's signed-rank test, Wilcoxon's rank-sum test, two-tailed paired and unpaired $t$ test, one-way analysis of variance (ANOVA), two-way ANOVA, Mann–Whitney $U$ test, and $\chi^2$ test. Results of statistical tests were summarized in Table S1. Statistical significance was indicated as follows: *$P < 0.05$, **$P < 0.01$, ***$P < 0.001$, and ****$P < 0.0001$. See the figure legends.

**Reporting summary**. Further information on research design is available in the Nature Research Reporting Summary linked to this article.

## Data availability
All data supporting the findings of this study are provided within the paper and its Supplementary information. All additional information will be made available upon reasonable request to the authors. Source data are provided with this paper.

## Code availability
Custom codes used for this study are uploaded to: https://github.com/xinkuaninGit/PAG_hunting.git.

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

## Acknowledgements

We thank Drs. Yunyun Han (Huazhong University of Science and Technology) and Mario Penzo (NIMH) for critical reading and editing of the manuscript and members of the Li laboratory for helpful discussions. This study was supported by the National Key Research and Development Program of China (2019YFA0801900), National Natural Science Foundation of China (31671105, 91857104, and 31771169), Major project of National Natural Science Foundation of China (61890953), Innovation Fund of Wuhan National Laboratory of Optoelectronics (WNLO), CAMS Innovation Fund for Medical Sciences (2019-I2M-5-057), and Shenzhen-Hong Kong Institute of Brain Science-Shenzhen Fundamental Research Institutions (NYKFKT20190017). We thank the Wuhan National Laboratory of Optoelectronics for providing an experimental platform. We thank X. Li and the Molecular Imaging Core Facility (MICF) of School of Life Science and Technology, ShanghaiTech University for microscopic imaging; Y. Xiong and the Molecular Cellular Core for slices and staining; J. Wang and Animal Room of Shanghai University of Science and Technology; the staff members of the Animal Facility at the National Facility for Protein Science in Shanghai (NFPS), Zhangjiang Lab, China for providing technical support and assistance.

## Author contributions

H.Y., X.X. and Z.C. performed most of the experiments. H.Y. performed in vivo single-unit and electromyogram electrode recordings. H.Y., Z.C. and X.X. performed behavioral manipulation experiments and evaluations. X.X. and J.D. wrote the most MATLAB code. Xu W. performed slice physiological recordings. H.Y., Xinxin W., P.H. and Z.-d.Z. performed immunohistochemistry. H.L. and W.L.S designed and supervised the study. H.Y., X.X., W.L.S. and H.L. wrote the manuscript.

## Competing interests

The authors declare no competing interests.
