## [Peer Review File · Nature Communications]

Periaqueductal Gray Neurons Encode the Sequential Motor Program in Hunting Behavior of MiceREVIEWER COMMENTS

Reviewer #1 (Remarks to the Author):

By employing in vivo optrode recordings and genetically encoded circuit analysis tools, the authors report that lateral periaqueductal gray (LPAG) neuronal activities are organized in a sequential manner to support the serial structure of hunting motor actions—cricket prey search, chase and attack (but not eating)—in mice. Specifically, the GABAergic and glutamatergic inputs from central amygdala (CeA), lateral hypothalamus (LH) and zona incerta (ZI) regulated the LPAG neuronal activity; i.e., LPAGVgat (vesicular GABA transporter) neurons were found to be involved in the prey detection, chase and attack phases of hunting behaviors, while LPAGVglu2 (vesicular glutamate transporter) neurons were only activated during the attack behavior. While this study is an extension of an earlier study that found that the SC (superior colliculus) ZIPAG circuit drives the complex hunting behavior in mice, the aforementioned findings provide significant new information. Below I listed several comments for the authors to consider.

1. Whether the animal's movement affected LPAG spike firing should be analyzed. Additional data such as velocity curves during each phase and the number of chases during the attack phase will provide better understanding of the roles of LPAG.
2. The average firing rates during several seconds (based on Fig. 1d; introduction, 1-2 s; chase, 2-3 s; attack, 13-14 s; eating, > 20s) likely contain mixed information. In addition to the coarse analyses, more narrow time-bin (< 1s) analyses involving the animal's speed will be needed. Also, it is unclear what is meant by 5 bins surrounding the peak value are unclear. The bin size should be clearly stated.
3. Do the PCA-clustered cells (Fig 1g, h) correspond to the peak response-aligned cell groups (the three columns in Fig 1i)? And the implications of each cluster type (e.g., what are the differences between type I vs. VI, III vs. VI, and II vs V) should be provided.
4. Whether the firing rate data between trials (a total of 5 trials) are reliable should be shown in the raster plots. All example rasters show only one trial.
5. HFD and object conditions might be good controls, but again, the animal's movement speed (and motivation level) in these conditions might be different from that in the cricket condition.
6. When LPAGVgat neurons were photo-inhibited, the latency to attack increased in the ArchT group (Fig 3c). Since the average latency was longer than photo-inhibition duration (30-s on), this means the animals were not successful in attacking the prey while the light was on in most cases. Whether the light stimulation affected the animal's movement per se will need to be assessed. If the light stimulation suppressed the animal's movement, then an alternative explanation is warranted.
7. The statistics for determining reduction and increase following input-specific lesions (Fig 5g-i) seem absent.
8. Describe the method (automated or blind observation?) for time-stamping specific behavioral phases to neuronal firing patterns.
9. The discussion on the releasing signal and the predatory motor sequence is confusing. Was this actually tested by removing the prey once the initial hunting action was triggered to see if the entirely behavioral sequence was completed?

Reviewer #2 (Remarks to the Author):

To whom it may concern,

please find below my comments on the manuscript 'Periaqueductal Neurons Encode the Sequential Motor Program in Hunting Behavior' by Yu et al.

The question of how behavioural sequences are encoded in the brain is a fundamental question in neuroscience, with a previous focus on learned tasks, such as go/no go tasks.

The study of instinctive behaviours has seen a recent revival. Instinctive behaviours are the natural repertoire of behaviours that evolution has shaped. They are both very reproducible yet also flexible because an animal's survival crucially depends upon them. This makes them highly suited to addressing questions of how neural mechanisms drive behaviour. Further, because they are highly conserved across the animal kingdom, by understanding the neural basis of instinctive behaviours, we will most certainly uncover fundamental principles of neural computations.

Most instinctive behaviours are not, as often wrongfully thought, stereotyped reflexes, but can contain long, complex and flexible action sequences. Thus, the question of how the brain encodes this, is a timely and important one.

Hunting in mice is a well-established instinctive behaviour with distinct behavioural phases, which can be readily elicited in a laboratory setting (recent studies: e.g. Han et al 2017, Park et al 2018, Li et al 2018, Hoy et al 2016) and is thus well-suited to tackling this question.

A large body of research has focused on the neural mechanisms underlying the initiation of instinctive behaviours, yet fewer studies have focused on how the sequential motor program is encoded.

Thus, the detailed description of the neuronal correlates in the lateral PAG of hunting along the entire action sequence is an important contribution to the field and warrants publication in Nature Communications.

General comments/ conceptual remarks:

- The writing appears at times inattentive, and the use of certain definitions and terms should be carefully reviewed by the authors.

- Despite a focus on the initiation of instinctive behaviours (especially hunting and escape), a significant number of studies have looked at the neural correlates of different behavioural 'timepoints' along the behavioural sequence (see papers above and e.g. Mearns et al 2020). The authors should discuss and acknowledge these studies, and change this in the introduction (line 51) accordingly, where they state that 'the sequential encoding of motor programs is mostly unknown'.

- I suggest the authors change the title of the study to 'Periaqueductal Gray Neurons Encode the Sequential Motor Program in Hunting Behavior'.

- line 27: 'cognitive and memory tasks' – memory is a cognitive process | similarly: line 68 'Therefore, like in cognitive tasks, [...] – what is the authors definition of cognition? Hunting behaviour most certainly entails cognitive processes, such as e.g. attention, decision making, learning and working memory (especially as in laboratory mice, cricket hunting has a strong learned component, as mice need a few training sessions to be able to hunt crickets successfully).

- Suggestion to replace 'innate' with 'instinctive'. In its commonly used definition, 'Innate' defines a behaviour that is executed without any prior learning, however the term 'instinctive' takes into account that most innate behaviours have learned elements. This is especially true for hunting in (laboratory) rodents, where most animals have an innate drive to hunt, but need practice to get good at it and do it efficiently (see also comment above). In this study, as in all other rodent studies, the animals are cricket-trained before the actual experiment takes place. I would thus recommend to replace the word 'innate' with 'instinctive' throughout the manuscript to avoid confusion.

- Line 52/53: 'Many brain regions have been associated with predatory hunting, including the superior colliculus, the amygdala, the hypothalamus and the brain stem' – according to most atlases, the superior colliculus is part of the midbrain, which is the topmost part of the brainstem, so listing the SC in addition to the brain stem is confusing, and the authors should state which other parts of the brainstem they are referring to.

- Line 60/61: '[...] with multiple functions, including defense, sensorimotor transformation, and motivated behaviours' – this list mixes up functions on very different levels. E.g. the behavioural output defense is both a sensorimotor transformation as well as a motivated behaviour. The authors should be more clear in what they want to convey conceptually.

- A general comment referring to both the first point on cognition, as well as to hunting being a motivated behaviour: In their final paragraph, the authors contrast instinctive behaviours with flexible, cognitive processes. They argue that instinctive behaviours are neither flexible nor cognitive, but stereotyped. This per se is not true. Hunting, at least in the wild, is a tremendously variable and nuanced behaviour. Importantly, decision-making in midbrain structures has been convincingly shown to exist in many species, clearly arguing for the presence of decision-making variables in the brainstem. Further, neural signatures of motivated and e.g. assessment behaviour have been described in the brainstem, again arguing against a purely 'stereotypical' output station. I suggest that the authors could broaden their discussion of this issue by, for example, discussing whether they think that flexibility in hunting would merely be conveyed through 'top-down' input from cortical areas (but see e.g. a recent pre-print by the lab of Karel Svoboda, Inagaki et al 2020, and literature on brainstem plasticity).

- The strong point of this paper is the finding that different populations of IPAG neurons are sequentially recruited in different phases of cricket hunting. The authors could expand on this finding much more with a more detailed analysis of the neural activity profile during these phases (as for example reliability over time, correlation to speed, vigour, repeated chases and attacks, etc). This is a wonderful dataset that is unfortunately very condensed in its presentation in Figure 1.

Comments on results/ figures:

Figure 1

- It would be good to analyse and show the relative increase/ decrease in firing of individual neurons in comparison to baseline firing. Do neurons with high/low baseline firing rates respond differentially during the different phases of hunting? Can the authors distinguish different types of neurons based on their action potential shape or are they, as shown by e.g. Deng et al 2016 in the dorsal PAG, not separable?

- What is the change in firing rate of neurons during failed chases, and failed attacks? – unsuccessful hunts? It would be very informative to analyse and display different sequences of a hunt in regards to the reliability of firing pattern of classified cells over time. Are introduction, chase and attack cells always and reliably active during consecutive chases? A hunt will usually contain multiple chase and attack phases, the authors should present a quantification of this in Fig 1.

- What do the authors think happens during the 'introduction' phase, could they give a bit more detail what the mouse is doing in this time? Could these be assessment cells as shown to exist by Deng et al 2016 and Masferrer et al 2018 in the dPAG?

- A quantification of firing rates of neurons over time without cricket insertion could be a useful comparison and to visualise the stability of firing rates over time.

- Fig 1i - Did the authors cross-validate the sorting of the heatmaps?

- Are the firing rates correlated to speed or head velocity? Rather than just showing average and normalised firing rates across the entire selected phase, the authors should analyse the correlation of firing rates to vigour, as it has been shown in many previous papers that PAG activity is tightly correlated to these parameters (see eg Deng et al 2016, Masferrer et al 2018, Evans et al 2018).

Figure 2:

- Is there a difference in baseline firing rates between VGAT and VGlut2 neurons? Is there a difference in absolute peak firing rates?

Figure 3:

- Histological examples of brain sections with ArchT spread/expression in the IPAG would be helpful to have in the supplemental figures. How many cells were infected and was the spread localised to the IPAG?

- Similar to the question regarding Fig 1, what was the effect of ArchT-mediated inhibition of VGAT neurons on speed? Could the reduction of efficiency be secondary to slowing the mouse down?

Figure 4:

- The spread of ArchT seems to go into the superior colliculus? Please quantify and provide clearer sections.

- Does the inhibition of VGAT neurons affect speed (both during baseline and during the chase)?

Figure 5 and Figure 6:

- The caspase experiments, where cells in IPAG-projecting cells in CeA, LH and ZI are complex experiments. However, without the corresponding circuit analysis of neural connectivity it is hard to interpret this data and no solid conclusions can be drawn. For example, one of many pitfalls of this experiment, as the authors state themselves, is the recurrent connectivity within the IPAG. It would have been more informative to record the activity of those brain areas during hunting, and acutely inactivate their inputs to the IPAG during the different phases of hunting. Have the authors performed these experiments and seen any effects? Further, and this goes for most behavioural experiments, supplemental figures with analysis and controls of baseline locomotion would be very useful. E.g. how do ablations of CeA, LH and ZI affect other non-hunting behaviours of mice? The authors should present controls to exclude that their results could be secondary to general effects of cell ablation in the brain areas.

Extended Data Figure 4

- The idea that VGAT IPAG neurons encode motivation is very interesting. However, based on the data the authors present, two problems arise as their two tests are based solely on ChR2 stimulation. First, given the pitfalls of ChR2 stimulation, this interpretation, which has important consequences in the conceptual analysis of PAG function, lacks controls and further experimental support (such as neural activity recordings of VGAT neurons during motivated behaviours). Second, VGAT neurons project to a vast number of other brain areas and importantly also inhibit local VGlut2 neurons that project to many other places including the lateral hypothalamic area (LHA). This glutamatergic IPAG-LHA projection has recently been suggested to provide an 'emotional drive for prey hunting' (Marín-Blasco et al 2021, Biorxiv). Thus, this finding could be secondary to a ChR2-induced inhibition of local VGlut2 neurons, or inactivation/indirect activation of other downstream targets. Thus, the authors may want to consider to present additional data to support this point or discuss the pitfalls and weaknesses of this experiment.

- The authors should show examples of at least 2 brains of the extend of VGAT:ChR2 infection within the PAG. As they are suggesting that IPAG neurons encode positive motivation, it is important to quantify the spread of viral infection and show that they haven't hit neighbouring brain areas such as vIPAG/dorsal raphe or superior colliculus.

Extended Data Figure 5:

- The authors should cite a recent study by the Lin lab (Falkner et al 2020), who have shown that IPAG VGlut2 neurons are also active during attack action in social aggression, with simultaneous recordings of jaw EMG and IPAG spiking during attack episodes (Fig 4 of that paper).

Minor comments:

- Check manuscript for grammar mistakes, e.g. line 80: replace LAG with LPAG, line 91: replace shuffled with shuffling, line 92: similar instead of approximate?, line 121: similar LPAG neurons – a similar percentage of LPAG neurons ...

Reviewer #3 (Remarks to the Author):

In this manuscript by Yu and Li, et al., the authors investigate the encoding of predatory behavioral sequences in GABAergic and glutamatergic lateral periaqueductal gray neurons. The authors isolated distinct encoding of the behavioral sequences by LPAG-GABA neuron subpopulations during prey detection, chase, and attack, and the contribution of LPAG-glutamate neurons to prey attack. The authors further demonstrate the necessity of these distinct populations to these behavioral sequences using temporally resolved inhibition. Overall, the manuscript is a technical achievement, integrating neural physiology, behavior, and circuit function and represents a significant advance in understanding the neural circuit dynamics underlying predatory behavior. I have only a few suggestions for strengthening the overall manuscript.

Specific comments:

1. The authors nicely demonstrate that inhibition of LPAG-Vglut2 neurons does not affect free reward consumption. However, their analysis of this population is much more superficial than the analysis of the LPAG-Vgat population. A more thorough analysis to juxtapose to the LPAG-Vgat population is warranted.
2. It is not clear why the data in figure 6 is normalized to the EYFP control. The authors should provide the EYFP data and do conventional statistical analysis of the input population inactivation compared to the controls and between groups.
3. The circuit level analysis represented in Figure 6h is incomplete. The authors do not investigate how the different inputs to the LPAG regulate the LPAG-Vglut2 population. It is not necessary to provide a full analysis of the role of these inputs in regulating the in vivo firing properties of LPAG-Vglut2 neurons during predatory behavior; however, the authors should perform optogenetic analysis of connectivity using slice electrophysiology. This can be easily achieved by injecting AAV-FLEX-ChR2-EYFP into the CeA, LH, or ZI of Vgat-Cre mice and AAV-FLEX-mCherry into the LPAG. This will isolate GABA neurons in the LPAG, and the authors can record labeled and unlabeled neurons (presumably glutamate) neurons.
4. The resolution of how GABA inputs to the LPAG regulate the activation of GABA neurons during hunting remains unclear. The authors propose disinhibitory mechanisms, but this is relatively unsatisfying. The authors should provide an analysis of the projection density and anatomical resolution of the different inputs to the LPAG to establish if there is any anatomical segregation of these inputs at the level of the LPAG.

Point-by-point responses to the reviewers' comments:

REVIEWER COMMENTS

Reviewer #1 (Remarks to the Author):

By employing in vivo optrode recordings and genetically encoded circuit analysis tools, the authors report that lateral periaqueductal gray (LPAG) neuronal activities are organized in a sequential manner to support the serial structure of hunting motor actions—cricket prey search, chase and attack (but not eating)—in mice. Specifically, the GABAergic and glutamatergic inputs from central amygdala (CeA), lateral hypothalamus (LH) and zona incerta (ZI) regulated the LPAG neuronal activity; i.e., LPAGVgat (vesicular GABA transporter) neurons were found to be involved in the prey detection, chase and attack phases of hunting behaviors, while LPAGVglut2 (vesicular glutamate transporter) neurons were only activated during the attack behavior. While this study is an extension of an earlier study that found that the SC (superior colliculus) ZIPAG circuit drives the complex hunting behavior in mice, the aforementioned findings provide significant new information. Below I listed several comments for the authors to consider.

1. Whether the animal's movement affected LPAG spike firing should be analyzed. Additional data such as velocity curves during each phase and the number of chases during the attack phase will provide better understanding of the roles of LPAG.

R: Thanks for these constructive comments. We performed the analyses of velocity curves during each phase and found that a subset of LPAG neurons activated during different hunting phases were also sensitive to the movement (**Supplementary Fig. 2g**). Correlation analysis revealed that the activity of these movement-activated neurons during the prey chase, but not during prey introduction, attack and eating phases, was linearly correlated with the movement speed. We quantified the number of chase rounds during the attack phase (1.44 ± 0.09) (**Supplementary Fig. 2c**).

c, The number of chase rounds during the attack phase ($n = 10$ mice). g, Top, normalized firing rates (black) and velocity curves (orange) during each hunting phase. Movement cells were defined that average firing rates during the movement were significantly higher than that of the station. Pie charts represent the overlap between movement cells and introduction cells, chase cells, attack cells or eating cells, respectively. Dotted lines aligned put in, chase onset, first attack and eating onset, respectively. Bottom, the activity-speed correlation ($P = 0.008$ during the

introduction, $P < 0.0001$ during the chase, $P = 0.5973$ during the attack, $P = 0.6652$ during the eating, Pearson's r). Data are shown as the mean \pm s.e.m.

2. The average firing rates during several seconds (based on Fig. 1d; introduction, 1-2 s; chase, 2-3 s; attack, 13-14 s; eating, > 20s) likely contain mixed information. In addition to the coarse analyses, more narrow time-bin (< 1s) analyses involving the animal's speed will be needed. Also, it is unclear what is meant by 5 bins surrounding the peak value are unclear. The bin size should be clearly stated.

R: To investigate whether the movement affected the activity of LPAG neurons across hunting, we re-recorded LPAG neurons during the cricket hunting task. We performed the analyses of velocity curves during hunting, and found that the activity of LPAG neurons was correlated with the movement speed. Each peak value in the attack phase represents one attack segment during the attack phase. The bin size is 0.5 s.

This new analysis is presented in the new **Fig. 1f**.

f, A typical neuronal activity trace and velocity curve during different predatory phases. Each green vertical line represents one attack segment. Dotted lines represent the timepoints of put-in of crickets, chase onset, first attack, and eating onset, respectively.

We quantified ridge-to-background ratio of the unshuffled and shuffled data to test whether the ordering of data is an artifact of sequence. The relevant comparison between the ordered unshuffled data and the ordered shuffled data illustrates the effects of sorting the data by peak time. The ridge was defined following the previous report in a virtual-navigation decision task, where the ridge was defined as 20 bins (64 ms/bin) surrounding the peak for a virtual-navigation decision task, as shown in the previous study¹. For the hunting behavior, the phase changes were in seconds. Therefore, we defined the ridge as the average activity in five bins (200 ms/bin) surrounding the peak in our study. The ridge (red color) of a demo neuron is shown in the following figure.

Left, normalized responses of a demo neuron during predatory hunting, sorted by its peak of the responses. Right, shuffled data from the same neuron as in the left panel (based on 1000 shuffles). Red shadows represent ridges. Dotted lines represent the timepoints of put-in of crickets, chase

onset, first attack, and eating onset, respectively.

3. Do the PCA-clustered cells (Fig 1g, h) correspond to the peak response-aligned cell groups (the three columns in Fig 1i)? And the implications of each cluster type (e.g., what are the differences between type I vs. VI, III vs. VI, and II vs V) should be provided.

R: As shown in the new **Supplementary Fig. 2i**, we calculated the average peak response time of PCA-clustered cells (Fig 1g, h) during five predatory phases and found that the PCA-clustered cells corresponded to the peak response-aligned cell groups (the three columns in Fig 1i).

i, The average peak response time of PCA-clustered cells during five predatory phases. Dotted lines represent the timepoints of put-in of crickets, chase onset and first attack, respectively.

The differences between type I vs. IV, III vs. VI, and II vs. V were the following:

1. Both type I and type IV neurons preferentially fired to the prey introduction. Yet, type I neurons showed a much larger response than type IV neurons and maintained high activity during the chase and attack phases (introduction-related cells, **Fig. 1h**). In contrast, the activity of type IV neurons diminished after the prey introduction. Thus, we speculate that type I neurons might serve a role in motivation that drives the whole hunting process, while type IV neurons might only function in the prey detection.

2. Both type III and type VI neurons preferentially fired to the prey chase. Yet, Type III neurons showed a much larger response than type VI neurons and maintained high activity in the attack phase (chase-related cells, **Fig. 1h**). In contrast, the activity of type VI neurons diminished after the prey chase. Thus, we reason that type III neurons might be recruited for the movement speed and type VI neurons might be recruited selectively for the prey chase.

3. Both type II and type V neurons preferentially fired to the prey attack. Yet, type II neurons showed a much larger response than type V neurons. Both types maintained high activity in the chase phase (attack-related cells, **Fig. 1h**). Thus, we speculate that type II and type V neurons might encode different attack actions, respectively.

4. Whether the firing rate data between trials (a total of 5 trials) are reliable should be shown in the raster plots. All example rasters show only one trial.

R: Thanks for the suggestion. As shown in the new **Supplementary Fig. 2f**, we plotted raster and peri-stimulus time histogram of LPAG neurons in five consecutive hunting trials and confirmed that the firing rate data among trials were reliable.

f, A typical neuronal activity trace (5 trials/session) during different predatory phases. The black dotted line represents a 95% confidence interval for firing rates of the baseline. The grey dotted lines represent the timepoints of put-in of crickets. The red dotted lines in the left represent the timepoints of chase onset and red lines in the right represent chase phases. The green dotted lines in the left represent the timepoints of first attack and green lines in the right represent attack phases. The blue dotted lines in the left represent the timepoints of eating onset and blue lines in the right represent eating phases.

5. HFD and object conditions might be good controls, but again, the animal's movement speed (and motivation level) in these conditions might be different from that in the cricket condition.

R: Thanks for the reviewer's comments. As shown in the new **Supplementary Fig. 4c**, we found that mice ran significantly faster toward the crickets than that to HFD and objects. The analysis of the movement speed has been shown above already.

c, Chase speed during running to different targets (n = 3 mice, one-way ANOVA, Tukey's multiple comparisons test, * $P < 0.01$, $P > 0.05$, n.s., no significance). Data are shown as the mean \pm s.e.m.

6. When LPAG^{Vgat} neurons were photo-inhibited, the latency to attack increased in the ArchT group (Fig 3c). Since the average latency was longer than photo-inhibition duration (30-s on), this means the animals were not successful in attacking the prey while the light was on in most cases. Whether the light stimulation affected the animal's movement per se will need to be assessed. If the light stimulation suppressed the animal's movement, then an alternative explanation is warranted.hn

R: To assess whether the light stimulation affected the animal's movement, LPAG^{Vgat-ArchT/EYFP} mice were placed on an acrylic arena (30 cm \times 30 cm \times 25 cm) and tested once in this arena. The test lasted 15 min and included 5-min laser off – 5-min laser on - 5min-laser off. During the laser stimulation, a 532-nm laser stimulation was delivered in 30-s on, followed by 30-s off cycles. We found that the photoinhibition via ArchT did not significantly affect the spontaneous movement of LPAG^{Vgat} mice.

These data are presented in the new **Supplementary Fig. 6g-i**.

g-i, The open field test. Scheme (**g**), total distance (**h**), average velocity (**i**) before, during and after bilateral inhibition of LPAG^{Vgat} neurons (n = 6 mice for each group, two-way repeated measures ANOVA, Bonferroni's multiple comparisons test, $P > 0.05$, n.s., no significance). Laser pattern: 532 nm, 12 mW, 30-s on/30-s off cycles. Data are shown as the mean \pm s.e.m.

7. The statistics for determining reduction and increase following input-specific lesions (Fig 5g-i) seem absent.

R: Thanks for the reviewer's reminder. We conducted statistical analysis by using the chi-square test. The significant difference was indicated in the new figure 5.

g-i, The percentage of cell types after ablation of GABAergic input neurons from the CeA (**g**), the LH (**h**), and the ZI (**i**), respectively (chi-square test, $*P < 0.05$, $**P < 0.01$, $***P < 0.001$, $****P < 0.0001$, $P > 0.05$, n.s., no significance).

8. Describe the method (automated or blind observation?) for time-stamping specific behavioral phases to neuronal firing patterns.

R: In our study, we used double-blind observation for time-stamping specific behavioral phases to neuronal firing patterns. We described it in the methods.

9. The discussion on the releasing signal and the predatory motor sequence is confusing. Was this actually tested by removing the prey once the initial hunting action was triggered to see if the entirely behavioral sequence was completed?

R: Thanks for raising this interesting concern. As suggested by reviewer #2 and classical reports^{2,3}, the predatory motor sequence is not only stereotypical under the same set of conditions but flexible among treatments or alternative stimuli. The stereotypical predatory motor sequence is released in response to a "releasing signal". Yet, the completion of the predatory sequence requires the continuous presence of prey. Once the prey is removed, the hunting behavior will stop (**Supplementary Movies 1 and 2**). These data are consistent with the previous study that prey

capture sequences in zebrafish are disrupted when we interfere with visual processing using genetic mutants or remove visual cues after the behavior is initiated⁴. Thus, the predatory action sequence is not only stereotypical with the continuous presence of prey but also flexible after altered sensory stimuli. We now clearly stated these points in the manuscript. (Lines 462-467)

Reviewer #2 (Remarks to the Author):

To whom it may concern,

Please find below my comments on the manuscript ‘Periaqueductal Neurons Encode the Sequential Motor Program in Hunting Behavior’ by Yu et al.

The question of how behavioural sequences are encoded in the brain is a fundamental question in neuroscience, with a previous focus on learned tasks, such as go/no go tasks.

The study of instinctive behaviours has seen a recent revival. Instinctive behaviours are the natural repertoire of behaviours that evolution has shaped. They are both very reproducible yet also flexible because an animal’s survival crucially depends upon them. This makes them highly suited to addressing questions of how neural mechanisms drive behaviour. Further, because they are highly conserved across the animal kingdom, by understanding the neural basis of instinctive behaviours, we will most certainly uncover fundamental principles of neural computations.

Most instinctive behaviours are not, as often wrongfully thought, stereotyped reflexes, but can contain long, complex and flexible action sequences. Thus, the question of how the brain encodes this, is a timely and important one.

Hunting in mice is a well-established instinctive behaviour with distinct behavioural phases, which can be readily elicited in a laboratory setting (recent studies: e.g. Han et al 2017, Park et al 2018, Li et al 2018, Hoy et al 2016) and is thus well-suited to tackling this question.

A large body of research has focused on the neural mechanisms underlying the initiation of instinctive behaviours, yet fewer studies have focused on how the sequential motor program is encoded.

Thus, the detailed description of the neuronal correlates in the lateral PAG of hunting along the entire action sequence is an important contribution to the field and warrants publication in Nature Communications.

General comments/ conceptual remarks:

- The writing appears at times inattentive, and the use of certain definitions and terms should be carefully reviewed by the authors.

R: We thanked the reviewer’s comments and had carefully checked and corrected the terms. For example, we have now replaced the word ‘innate’ with ‘instinctive’ as suggested.

- Despite a focus on the initiation of instinctive behaviours (especially hunting and escape), a significant number of studies have looked at the neural correlates of different behavioural ‘timepoints’ along the behavioural sequence (see papers above and e.g. Mearns et al 2020). The authors should discuss and acknowledge these studies, and change this in the introduction (line 51) accordingly, where they state that ‘the sequential encoding of motor programs is mostly unknown’.

R: Thanks for the reminder. We have cited these paper as the following: “It is a well-established instinctive behavior with a specific sequential behavior actions, including the prey search, pursuit, attack, and consumption, which can be readily elicited in a laboratory setting⁴⁻⁶. These behavioral actions are very stereotypical under the same conditions but flexible among treatments or alternative stimuli^{2,3}, making them highly suitable for addressing the fundamental question of how neuronal activity drives complex behavior. Further, since the hunting behavior is highly conserved, understanding its neural basis is likely uncover the basic principles of behavioral organization. The sequential neuronal activity has been established in the hippocampus and motor cortex during memory or decision-making tasks⁷⁻¹¹, which is essential for navigation planning⁷ and motor generation¹¹. However, the neural substrate underlying instinctive behavioral sequences such as that in hunting, is still mostly unknown”. (Lines 43-52)

- I suggest the authors change the title of the study to ‘Periaqueductal Gray Neurons Encode the Sequential Motor Program in Hunting Behavior’.

R: We have now changed the title to ‘Periaqueductal Gray Neurons Encode the Sequential Motor Program in Hunting Behavior’.

- line 27: ‘cognitive and memory tasks’ – memory is a cognitive process | similarly: line 68 ‘Therefore, like in cognitive tasks, [...] – what is the authors definition of cognition? Hunting behaviour most certainly entails cognitive processes, such as e.g. attention, decision making, learning and working memory (especially as in laboratory mice, cricket hunting has a strong learned component, as mice need a few training sessions to be able to hunt crickets successfully).

R: Thanks for this nice comment. We agreed with the reviewer and have changed to ‘Sequential encoding of motor programs is essential for behavior generation.’ (Line 25) and ‘Therefore, like in the hippocampus and motor cortex, we hypothesized that LPAG neuronal activities are organized in a sequential framework to support the serial structure of hunting motor actions.’ (Lines 67-69)

- Suggestion to replace ‘innate’ with ‘instinctive’. In its commonly used definition, ‘Innate’ defines a behaviour that is executed without any prior learning, however the term ‘instinctive’ takes into account that most innate behaviours have learned elements. This is especially true for hunting in (laboratory) rodents, where most animals have an innate drive to hunt, but need practice to get good at it and do it efficiently (see also comment above). In this study, as in all other rodent studies, the animals are cricket-trained before the actual experiment takes place. I would thus recommend to replace the word ‘innate’ with ‘instinctive’ throughout the manuscript to avoid confusion.

R: We acknowledged this comment and have replaced the word ‘innate’ with ‘instinctive’ throughout the manuscript.

- Line 52/53: ‘Many brain regions have been associated with predatory hunting, including the superior colliculus, the amygdala, the hypothalamus and the brain stem’ – according to most atlases, the superior colliculus is part of the midbrain, which is the topmost part of the brainstem, so listing the SC in addition to the brain stem is confusing, and the authors should state which other parts of the brainstem they are referring to.

R: We have replaced the “the brain stem” with “the periaqueductal gray”. (Line 54)

- Line 60/61: ‘[...] with multiple functions, including defense, sensorimotor transformation, and motivated behaviours’ – this list mixes up functions on very different levels. E.g. the behavioural output defense is both a sensorimotor transformation as well as a motivated behaviour. The authors should be more clear in what they want to convey conceptually.

R: We have now corrected that the PAG is a midbrain structure with multiple functions, including defensive¹²⁻¹⁴, social¹⁵, maternal¹⁶ and emotional behaviors¹⁷. (Lines 61-62)

- A general comment referring to both the first point on cognition, as well as to hunting being a motivated behaviour: In their final paragraph, the authors contrast instinctive behaviours with flexible, cognitive processes. They argue that instinctive behaviours are neither flexible nor cognitive, but stereotyped. This per se is not true. Hunting, at least in the wild, is a tremendously variable and nuanced behaviour. Importantly, decision-making in midbrain structures has been convincingly shown to exist in many species, clearly arguing for the presence of decision-making variables in the brainstem. Further, neural signatures of motivated and e.g. assessment behaviour have been described in the brainstem, again arguing against a purely ‘stereotypical’ output station. I suggest that the authors could broaden their discussion of this issue by, for example, discussing whether they think that flexibility in hunting would merely be conveyed through ‘top-down’ input from cortical areas (but see e.g. a recent pre-print by the lab of Karel Svoboda, Inagaki et al 2020, and literature on brainstem plasticity).

R: Thanks a lot for the constructive advice. We have now corrected the final paragraph as the following: “In summary, the central nervous system provides stereotypical but flexible functional neuronal ensembles to encode instinctive behaviors. When an instinctive behavior generates, a series of neuronal ensembles are activated sequentially, which chains various actions into sequences. The actions are considered to be stereotyped, where one neuronal ensemble activated may induce only one fixed type of action output. In contrast, the behavioral sequence is flexible, where the order of ensembles could be adjusted by earlier experience (motor planning, top-down) and sensory evidence (close-range interaction, bottom-up). In such cases, behavioral sequences may be hierarchically organized and influenced by internal states or sensory stimuli. We reason that in this manner, the behavior is robustly performed with low cost but still retains flexibility. Thus, we provide a framework of neuronal ensemble sequences across predatory hunting, which may shed light on other instinctive behaviors.” (Lines 468-478)

- The strong point of this paper is the finding that different populations of IPAG neurons are sequentially recruited in different phases of cricket hunting. The authors could expand on this finding much more with a more detailed analysis of the neural activity profile during these phases (as for example reliability over time, correlation to speed, vigour, repeated chases and attacks, etc).

This is a wonderful dataset that is unfortunately very condensed in its presentation in Figure 1.

R: Thanks for the suggestions. Also mentioned by reviewer #1, we have included the reliability of neuronal firing with repeated trials (**Supplementary Fig. 2f**) and analyzed the correlation between the neuronal firing and movement speed (**Supplementary Fig. 2g**). According to previous studies^{13,18}, we used the movement speed as an indicator of vigor and analyzed the correlation between the movement speed and neuronal firing during each phase (**Supplementary Fig. 2g**). Also, we quantified repeated chases (**Supplementary Fig. 2a**) and attacks (**Supplementary Fig. 2b**).

f, A typical neuronal activity trace (5 trials/session) during different predatory phases. The black dotted line represents a 95% confidence interval for firing rates of the baseline. The grey dotted lines represent the timepoints of put-in of crickets. The red dotted lines in the left represent the timepoints of chase onset and red lines in the right represent chase phases. The green dotted lines in the left represent the timepoints of first attack and green lines in the right represent attack phases. The blue dotted lines in the left represent the timepoints of eating onset and blue lines in the right represent eating phases.

g, Top, normalized firing rates (black) and velocity curves (orange) during each hunting phase. Movement-specific cells were defined that average firing rates during the movement were significantly higher than that of the station. Pie charts represent the overlap between movement cells and introduction cells, chase cells, attack cells or eating cells, respectively. Dotted lines aligned the timepoints of put-in of crickets, chase onset, first attack and eating onset, respectively. Bottom, the activity–speed correlation ($P = 0.008$ during the introduction, $P < 0.0001$ during the chase, $P = 0.5973$ during the attack, $P = 0.6652$ during the eating, Pearson's r).

a, The number of chase rounds (n = 10 mice). **b**, The number of attack segments (n = 10 mice).

Comments on results/ figures:

Figure1

- It would be good to analyze and show the relative increase/decrease in firing of individual neurons in comparison to baseline firing.

R: Thanks for this suggestion. We analyzed the relative increase/decrease in firing rates of individual neurons in comparison to baseline firing rates and found that the change in firing rates of individual neurons during the introduction, chase, or attack phase was not correlated with the baseline, while firing rates of the eating phase had a weak negative correlation with the baseline.

These data are presented in the new **Supplementary Fig. 5i-l**.

i, Correlation between the change in firing rates during the introduction phase and the baseline firing rates ($P = 0.442$, Pearson's r). **j**, Correlation between the change in firing rates during the chase phase and the baseline firing rates ($P = 0.997$, Pearson's r). **k**, Correlation between the change in firing rates during the attack phase and the baseline firing rates ($P < 0.0001$, Pearson's r). **l**, Correlation between the change in firing rates during the eating phase and the baseline firing rates ($P < 0.0001$, Pearson's r).

Do neurons with high/low baseline firing rates respond differentially during?

R: We compared the percentage of responsive cells with high/low baseline firing rates during different predatory phases and found that there was no significant difference between both the types. The neurons with low baseline firing rates were defined as firing at a lower than the average baseline. The neurons with high baseline firing rates were defined as firing at a higher than average baseline. The data is shown in the following figure.

The percentage of responsive cells with high/low baseline firing rates during different predatory phases (chi-square test, $P > 0.05$, n.s., no significance).

Can the authors distinguish different types of neurons based on their action potential shape or are they, as shown by e.g. Deng et al 2016 in the dorsal PAG, not separable?

R: We compared LPAG neurons, LPAG^{Vgat} neurons and LPAG^{Vglut2} neurons based on three electrophysiological properties, including firing rate, half-width, trough-to-peak as the previous study¹⁹. However, we found that the recorded neurons were not separable from each other. The data demonstrates that the sorting method based on the electrophysiological properties of neurons is not reliable in the LPAG.

These data are presented in the new **Supplementary Fig. 5g, h**.

g, The distribution of LPAG neurons, tagged LPAG^{Vgat} neurons and tagged LPAG^{Vglut2} neurons based on firing rate, half-width and trough-to-peak. **h**, Left, the distribution of LPAG neurons, tagged LPAG^{Vgat} neurons and tagged LPAG^{Vglut2} neurons based on half-width and trough-to-peak. Middle, the distribution of LPAG neurons, tagged LPAG^{Vgat} neurons and tagged LPAG^{Vglut2} neurons based on trough-to-peak and firing rate. Right, the distribution of LPAG neurons, tagged LPAG^{Vgat} neurons and tagged LPAG^{Vglut2} neurons based on half-width and firing rate.

- What is the change in firing rate of neurons during failed chases, and failed attacks? – unsuccessful hunts? It would be very informative to analyze and display different sequences of a hunt in regards to the reliability of firing pattern of classified cells over time. Are introduction,

chase and attack cells always and reliably active during consecutive chases? A hunt will usually contain multiple chase and attack phases, the authors should present a quantification of this in Fig 1.

R: We performed a more detailed analysis in Fig 1, and expanded on our findings. Firstly, we analyzed the change in firing rate of neurons during failed chases and failed attacks, and found that the average firing rates during failed chases and attacks were significantly higher than the baseline (**Supplementary Fig. 2d, e**). Secondly, we quantified chase rounds (1.433 ± 0.104 / session) and attack segments (1.416 ± 0.08 / session) (**Supplementary Fig. 2a, b**). Thirdly, we found that introduction, chase and attack cells were reliably active yet with variations during consecutive chases (**Supplementary Fig. 2f**).

These data are presented in the new **Supplementary Fig. 2a, b, d-f**.

a, The number of chase rounds ($n = 10$ mice). **b**, The number of attack segments ($n = 10$ mice). **d**, Firing rates of LPAG neurons during failed chases ($n = 151$ units, Wilcoxon signed-rank test, $****P < 0.0001$). **e**, Firing rates of LPAG neurons during failed attacks ($n = 155$ units, Wilcoxon signed-rank test, $****P < 0.0001$). **f**, A typical neuronal activity trace (5 trials/session) during different predatory phase. The black dotted line represents a 95% confidence interval for firing rates of the baseline. The grey dotted lines represent the timepoints of put-in of crickets. The red dotted lines in the left represent the timepoints of chase onset and red lines in the right represent chase phases. The green dotted lines in the left represent the timepoints of first attack and green lines in the right represent attack phases. The blue dotted lines in the left represent the timepoints of eating onset and blue lines in the right represent eating phases. Data are shown as the mean \pm s.e.m.

- What do the authors think happens during the ‘introduction’ phase, could they give a bit more detail what the mouse is doing in this time? Could these be assessment cells as shown to exist by Deng et al 2016 and Masferrer et al 2018 in the dPAG?

R: To further define the ‘introduction’ phase, we performed single-unit recordings in head-fixed mice, and delivered visual and auditory stimuli which occurred during cricket introduction (**Supplementary Fig. 3a**). We found that a subset of LPAG neurons were excited by the moving visual stimulus, while few recorded neurons were sensitive to the static visual spot or the auditory

stimulus (**Supplementary Fig. 3b, c**), suggesting LPAG neurons detect prey motion signals during their introduction phase. Moreover, mice need to assess whether the prey target was dangerous during the introduction phase. Therefore, we ran a risk-assessment assay by presenting an awake rat to the hunting mouse and simultaneously recorded single-units in the LPAG (**Supplementary Fig. 3d**). Interestingly, we found that 47.1% of assessment cells were introduction cells, while only a small percentage of introduction cells (8.8%) were risk-assessment cells (**Supplementary Fig. 3e-f**). Therefore, these results demonstrate that the introduction phase is a complicated behavioral process that includes sensory detection and risk assessment. These new data are presented in the new **Supplementary Fig. 3**.

a, Scheme of head-fixed experimental device for single-unit recording during visual and auditory stimuli. **b**, Top, heatmap of moving dot-induced neuronal activities by the black dot from ipsi. to contra. ($n = 267$ units). Bottom, normalized firing rates of visual-excited neurons by the moving dot in two directions. The number of visual-excited neurons from ipsi. to contra. was 111, while from ipsi. to contra. was 95. The dotted lines respectively aligned to the time point of 2 s-duration stable dot, 4 s-duration moving dot, 4 s-duration from moving to stable dot. **c**, Top, heatmap of tone-induced responses ($n = 267$ units). Bottom, normalized firing rates of 267 recorded neurons. Tone was delivered between the two dotted lines (60 dB, 0.5 s). **d**, Scheme of single-unit recording during the introduction phase and the risk assessment. **e**, Top, heatmap of assessment-excited neuronal activities ($n = 17$ units). Bottom, normalized firing rates of assessment cells. The dotted line aligned to the assessment onset. **f**, The overlap between introduction cells and assessment cells.

- A quantification of firing rates of neurons over time without cricket insertion could be a useful comparison and to visualize the stability of firing rates over time.

R: As mentioned above, we have shown the baseline firing rates across five successive trials in one session and visualized the stability of firing rates over time. The baseline was shown in light purple color. These data suggest that the firing rates over time are stable.

A typical neuronal activity trace across 5 successive trials of one session. The light purple shades represent the baseline. The black dotted line represents a 95% confidence interval for firing rates of the baseline. The grey dotted lines represent the timepoints of put-in of crickets. The red dotted lines represent the timepoints of chase onset. The green dotted lines represent the timepoints of first attack. The blue dotted lines represent the timepoints of eating onset.

- Fig 1i - Did the authors cross-validate the sorting of the heatmaps?

R: To perform cross-validation of the sorting of the heatmaps in Fig 1i, we randomly selected 50% and 30% of recorded cells and aligned them according to their peak firing rates, respectively. We found that the active phases of each neuron formed a sequence chain of neuronal activation spanning the entire predatory process and the active phases of another 50% of neurons produced nearly identical results (**Supplementary Fig. 2h**). These data indicate that this pattern is not an artifact of sorting. We did not directly compare the activity pattern of neurons across different hunting trials since we have shown the activity values (firing rates) during different hunting trials were different (**Supplementary Fig. 2f**).

The analysis is presented in the new **Supplementary Fig. 2h**.

h, Top, normalized responses of 50% (left) and another 50% (right) of recorded neurons during predatory hunting, sorted by their peak of responses. Bottom, normalized responses of 30% (left) and another 70% (right) of recorded neurons during predatory hunting, sorted by their peak of responses. Dotted lines represent the timepoints of put-in of crickets, chase onset, first attack, and eating onset, respectively.

- Are the firing rates correlated to speed or head velocity? Rather than just showing average and normalised firing rates across the entire selected phase, the authors should analyze the correlation of firing rates to vigor, as it has been shown in many previous papers that PAG activity

is tightly correlated to these parameters (see eg Deng et al 2016, Masferrer et al 2018, Evans et al 2018).

R: According to previous studies^{13,18}, we used the movement speed as an indicator of vigor and analyzed the correlation between the movement speed and neuronal firing during each phase. We found that a subset of LPAG neurons activated during different hunting phases were also sensitive to the movement. Correlation analysis revealed that the activity of these movement-activated neurons during the prey chase, but not during the prey introduction, attack and eating phases, was linearly correlated with the movement speed.

The data is presented in the new **Supplementary Fig. 2g**.

g, Top, normalized firing rates (black) and velocity curves (orange) during each hunting phase. Movement-specific cells were defined that average firing rates during the movement were significantly higher than that of the station. Pie charts represent the overlap between movement cells and introduction cells, chase cells, attack cells or eating cells, respectively. Dotted lines aligned the timepoints of put-in of crickets, chase onset, first attack and eating onset respectively. Bottom, the activity–speed correlation ($P = 0.008$ during the introduction, $P < 0.0001$ during the chase, $P = 0.5973$ during the attack, $P = 0.6652$ during the eating, Pearson's r).

Figure 2:

- Is there a difference in baseline firing rates between VGAT and VGlut2 neurons? Is there a difference in absolute peak firing rates?

R: Following the reviewer's suggestion, we analyzed the differences in baseline and absolute peak firing rates between LPAG^{Vgat} and LPAG^{Vglut2} neurons. We found that the baseline and peak firing rates of LPAG^{Vgat} neurons were significantly higher than those of LPAG^{Vglut2} neurons.

These data are presented in the new **Supplementary Fig. 5e, f**.

e, Baseline firing rates of all tagged LPAG^{Vgat} (light purple) and LPAG^{Vglut2} (light blue) neurons (Mann Whitney U test, $***P < 0.001$). **f**, Peak firing rates of all tagged LPAG^{Vgat} (light purple) and LPAG^{Vglut2} (light blue) neurons (Mann Whitney U test, $**P < 0.01$). Data are shown as the

mean \pm s.e.m.

Figure 3:

- Histological examples of brain sections with ArchT spread/expression in the IPAG would be helpful to have in the supplemental figures. How many cells were infected and was the spread localised to the IPAG?

R: Thanks a lot for the suggestion. We did the quantitative analysis and found that LPAG^{Vgat} neurons infected by ArchT were about 39.6% of total LPAG cells (**Supplementary Fig. 6a, b**). About 88.4% of Vgat⁺ neurons by ArchT infection were localized to the LPAG (**Supplementary Fig. 6f**), and the optical fibers were directly inserted above the LPAG, as shown in **Fig. 3a**. Therefore, our inhibition of LPAG^{Vgat} neurons was relatively selective.

a, Left, scheme for injections of Vgat-IRES-Cre mice. Right, histology of injection sites in the LPAG. Scale bars, 300 μm. **b**, Percentage of LPAG^{Vgat-ArchT}/LPAG^{DAPI} neurons (n = 4 mice). **f**, Percentage of Vgat⁺ neurons by ArchT infected spread localized to the LPAG.

- Similar to the question regarding Fig 1, what was the effect of ArchT-mediated inhibition of VGAT neurons on speed? Could the reduction of efficiency be secondary to slowing the mouse down?

R: We found that photoinhibition of LPAG^{Vgat} neurons during the chase phase did not significantly change the chase speed (**Supplementary Fig. 6c**). Nevertheless, to directly assess whether the light stimulation affected the animal's movement per se, LPAG^{Vgat-ArchT/EYFP} mice were placed on an acrylic arena (30 cm × 30 cm × 25 cm) and tested once in this arena (open field tests). The test lasted 15 min and included 5-min laser off – 5-min laser on – 5-min laser off. During the laser stimulation, a 532-nm laser stimulation was delivered in 30-s on, followed by 30-s off cycles. We found that photoinhibition LPAG^{Vgat} neurons via ArchT did not significantly affect the spontaneous movement of mice. So, the reduction of predatory efficiency is not likely to be due to the decrease in speed (**Supplementary Fig. 6g-i**).

These data are presented in the new **Extended Data Fig. 6c and 6g-i**.

c, Speed during optogenetic inhibition of LPAG^{Vgat} neurons in the chase phase (n = 6 mice each, two-tailed paired t-test, $P > 0.05$, n.s., no significance). **g-i**, The open field test. Scheme (**g**), total

distance (**h**), average velocity (**i**) before, during and after bilateral inhibition of LPAG^{Vgat} neurons (n = 6 mice for each group, two-way repeated measures ANOVA, Bonferroni's multiple comparisons test, $P > 0.05$, n.s., no significance). Laser pattern: 532 nm, 12 mW, 30-s on/30-s off cycles. Data are shown as the mean \pm s.e.m.

Figure 4:

- The spread of ArchT seems to go into the superior colliculus? Please quantify and provide clearer sections.

R: Thanks a lot. We now provide a clearer section in Fig. 4a. We also quantified LPAG^{Vglut2} neurons infected by ArchT, and found that the percentage of ArchT/DAPI was about 35.3% (**Supplementary Fig. 6d, e**). About 89.5% of Vglut2+ neurons by ArchT infection were localized to the LPAG (**Supplementary Fig. 6f**), and we did not notice any infection in the SC area. Also, the optical fibers were directly inserted above the LPAG, as shown in **Fig. 4a**. Therefore, the inhibition was relatively selective.

d, Left, scheme for injections of Vglut2-IRES-Cre mice. Right, histology of injection sites in the LPAG. Scale bars, 300 μ m. **e**, Percentage of LPAG^{Vglut2-ArchT}/LPAG^{DAPI} neurons (n = 4 mice). **f**, Percentage of Vglut2+ neurons by ArchT infected spread localized to the LPAG.

- Does the inhibition of VGAT neurons affect speed (both during baseline and during the chase)?

R: Please check the responses above. The speed during the spontaneous movement or the chase phase is not significantly affected by photoinhibition.

Figure 5 and Figure 6:

- The caspase experiments, where cells in IPAG-projecting cells in CeA, LH and ZI are complex experiments. However, without the corresponding circuit analysis of neural connectivity it is hard to interpret this data and no solid conclusions can be drawn. For example, one of many pitfalls of this experiment, as the authors state themselves, is the recurrent connectivity within the IPAG. It would have been more informative to record the activity of those brain areas during hunting, and acutely inactivate their inputs to the IPAG during the different phases of hunting. Have the authors performed these experiments and seen any effects? Further, and this goes for most behavioural experiments, supplemental figures with analysis and controls of baseline locomotion would be very useful. E.g. how do ablations of CeA, LH and ZI affect other non-hunting behaviours of mice? The authors should present controls to exclude that their results could be secondary to general effects of cell ablation in the brain areas.

R: Thanks for these helpful comments. Firstly, to directly assess connectivity of CeA/LH/ZI

neurons and LPAG neurons, we unilaterally injected AAV-hSyn-FLEX-EGFP-WPRE-hGFP into the LPAG of *Vgat-IRES-Cre* mice and injected AAV-hEF1a-DIO-hChR2 (H134R)-mCherry into the CeA, the LH, and the ZI, and recorded postsynaptic currents from *Vgat*⁺ and *Vgat*⁻ (presumably glutamatergic) neurons in the LPAG by optogenetically activating CeA/LH/ZI-LPAG projections, respectively. We found that *Vgat*⁺ and *Vgat*⁻ neurons recorded in the LPAG received monosynaptic inhibitory inputs from the CeA, the LH and the ZI. Interestingly, we found the ratios of LPAG GABAergic and glutamatergic neurons innervated by upstream GABAergic CeA/LH/ZI neurons were different. 42.9% of CeA^{*Vgat*}-projecting LPAG neurons were GABAergic and 30% of CeA^{*Vgat*}-projecting LPAG neurons were glutamatergic; 61.1% LH^{*Vgat*}-projecting LPAG neurons were GABAergic and 40% of LH^{*Vgat*}-projecting LPAG neurons were glutamatergic; while 42.9% ZI^{*Vgat*}-projecting LPAG neurons were GABAergic and 40.9% of ZI^{*Vgat*}-projecting LPAG neurons were glutamatergic. Altogether, these data indicate that the CeA, the LH, and the ZI are functionally connected with the LPAG, respectively.

These data are presented in the new **Supplementary Fig. 11**.

a-c, Connectivity of CeA-LPAG pathway. **a**, Experimental schematic. **b**, Example traces (left, by application of PTX) and quantitative analyses (right) of the amplitude of light-evoked IPSCs from the LPAG^{*Vgat*}⁺ or LPAG^{*Vgat*}⁻ neuron (two-tailed unpaired t-test, $P > 0.05$, n.s., no significance). **c**, Example traces (left, by application of TTX and 4-AP) and connection probabilities (right) of CeA-LPAG pathway. **d-f**, Connectivity of LH-LPAG pathway. **d**, Experimental schematic. **e**, Example traces (left, by application of PTX) and quantitative analyses (right) of the amplitude of light-evoked IPSCs from the LPAG^{*Vgat*}⁺ or LPAG^{*Vgat*}⁻ neuron (two-tailed unpaired t-test, $P > 0.05$, n.s., no significance). **f**, Example traces (left, by application of TTX and 4-AP) and connection probabilities (right) of LH-LPAG pathway. **g-i**, Connectivity of ZI-LPAG pathway. **g**, Experimental schematic. **h**, Example traces (left, by application of PTX) and quantitative analyses (right) of the amplitude of light-evoked IPSCs from the LPAG^{*Vgat*}⁺ or LPAG^{*Vgat*}⁻ neuron (two-tailed unpaired t-test, $P > 0.05$, n.s., no significance). **i**, Example traces (left, by application of TTX and 4-AP) and connection probabilities (right) of ZI-LPAG pathway. Data are shown as the mean \pm s.e.m.

For inhibition of upstream neurons, we would have performed the acute inhibition as suggested by the reviewer. Yet, photoinhibitions have some limitations for our experimental goal, such as limited laser coverage and variations in laser penetrance. Since the LH is large and the ZI is long, photoinhibition is less likely to cover the whole area. Therefore, after considering the limitations in photoinhibition, we decided to use taCasp3 to kill the entire upstream neurons and performed *in vivo* single-unit recordings on LPAG neurons after input-specific lesions.

To explore whether locomotion was affected after ablation of LPAG-projecting GABAergic neurons from the CeA/LH/ZI, we ran open field assay to test the animal's movement speed. From the spontaneous movement videos, we did not find any apparent deficit in locomotion (**Supplementary Movies 3-8**). To perform the OFT, LPAG^{Casp3/EYFP} mice were placed on an acrylic arena (30 cm × 30 cm × 25 cm) and tested once in this arena. The test lasted 5 min. We found that the locomotion activity was not significantly changed after ablation of LPAG-projecting GABAergic neurons from the CeA or the LH (**Supplementary Fig. 14a, b and f, g**). In contrast, the locomotion activity was reduced significantly after ablation of LPAG-projecting GABAergic neurons from the ZI (**Supplementary Fig. 14k, l**), which is consistent with our previous report that genetic deletion of ZI GABA reduces the movement speed²⁰.

These data are presented in the new **Supplementary Fig. 14a, b, f, g, k, l**.

a-b, Ablation of LPAG-projecting GABAergic neurons from the CeA. **a**, Experimental scheme. **b**, The open field test. Total distance (left), average velocity (right) ($n = 10$ mice, two-tailed unpaired t-test, $P > 0.05$, n.s., no significance). **f-g**, Ablation of LPAG-projecting GABAergic neurons from the LH. **f**, Experimental scheme. **g**, The open field test. Total distance (left), average velocity (right) ($n = 8$ mice, two-tailed unpaired t-test, $P > 0.05$, n.s., no significance). **k-l**, Ablation of LPAG-projecting GABAergic neurons from the ZI. **k**, Experimental scheme. **l**, The open field test. Total distance (left), average velocity (right) ($n = 10$ mice, two-tailed unpaired t-test, $**P < 0.01$). Data are shown as the mean \pm s.e.m.

Extended Data Figure 4

- The idea that VGAT IPAG neurons encode motivation is very interesting. However, based on the data the authors present, two problems arise as their two tests are based solely on ChR2 stimulation. First, given the pitfalls of ChR2 stimulation, this interpretation, which has important consequences in the conceptual analysis of PAG function, lacks controls and further experimental support (such as neural activity recordings of VGAT neurons during motivated behaviours). Second, VGAT neurons project to a vast number of other brain areas and importantly also inhibit local VGLUT2 neurons that project to many other places including the lateral hypothalamic area (LHA). This glutamatergic IPAG-LHA projection has recently been suggested to provide an ‘emotional drive for prey hunting’ (Marín-Blasco et al 2021, Biorxiv). Thus, this finding could be secondary to a ChR2-induced inhibition of local VGLUT2 neurons, or inactivation/indirect activation of other downstream targets. Thus, the authors may want to consider to present additional data to support this point or discuss the pitfalls and weaknesses of this experiment.

R: Thanks a lot for this wonderful advice. To test the hypothesis that LPAG^{Vgat} neurons encode the motivation, we used optrode recordings when mice performed an operant task to receive water reward. Head-fixed mice were water-restricted. Each trial started with a whisker stimulus (CS) delivered, and mice were trained to lick for 5- μ l water in a 3-s time window on a fixed ratio of one (FR1) and then gradually increased till FR = 8 (Fig. a). 20% of trials were intentionally set to be no reward (Fig. b). We recorded 139 neurons from 2 Vgat-IRES-Cre mice and 3 Vglut2-IRES-Cre mice, with 18 tagged Vgat⁺ neurons, and 16 tagged Vglut2⁺ neurons (Fig. c). Interestingly, we found that both Vgat⁺ and Vglut2⁺ neurons were activated by CS. Only the activity of Vgat⁺ neurons was sustained until the reward was obtained, suggesting that LPAG^{Vgat} neurons were involved in reward seeking or motivation (Fig. d-f). Together, these preliminary results indicate that LPAG^{Vgat} neurons might encode a positive motivational value. However, we agree with the reviewer that it is still possible that LPAG^{Vglut2} neurons might get inhibited by LPAG^{Vgat} neurons locally and target the downstream neurons in the LH (Marín-Blasco et al 2021, Biorxiv). Therefore, as we have not reached a solid conclusion, we did not include the Vgat⁺ single-unit recording data and wrote down the possibility that both types of neurons might encode motivation in the discussion. (Lines 404-409)

a, Top, schematic of a head-fixed experimental device for optrode recordings during reward seeking. Bottom, the behavioral paradigm of the conditioned stimulus. b, Average lick rate in reward seeking conditioning task (n = 6 mice, 32 sessions). The two red dotted lines represented

conditioned stimulus duration. The blue line represented reward, the orange line represented no reward and the gray line represented omission. **c**, Heatmap of the responses of all recorded neurons in reward-seeking task. 0–0.5 s represented conditioned stimulus, 0.5–2 s represented reward seeking. Light purple and light blue represented tagged $Vgat^+$ and $Vglut2^+$ neurons, respectively ($n = 139$ units). **d**, Z-score of tagged $Vgat^+$ and $Vglut2^+$ neurons response to sensation and reward seeking. **e**, Mean Z-score (0.5–2 s) of tagged $Vgat^+$ and $Vglut2^+$ neurons response to reward seeking (two-sided unpaired t test, $**P < 0.01$). **f**, Mean Z-score (0–0.5 s) of tagged $Vgat^+$ and $Vglut2^+$ neurons response to sensation (two-sided unpaired t test, $P > 0.05$, n.s., no significance). Data are shown as the mean \pm s.e.m.

- The authors should show examples of at least 2 brains of the extend of VGAT: Chr2 infection within the PAG. As they are suggesting that IPAG neurons encode positive motivation, it is important to quantify the spread of viral infection and show that they haven't hit neighbouring brain areas such as vIPAG/dorsal raphe or superior colliculus.

R: In RTPP and ICSS tests, we unilaterally injected Cre-inducible adeno-associated virus (AAV) expressing Chr2 into the LPAG of the $Vgat$ -IRES-Cre and implanted optical fibers bilaterally into the LPAG (**Supplementary Fig. 8a**). As shown in the following figures, the fiber was directly inserted above the LPAG (**Supplementary Fig. 8b**) and $LPAG^{Vgat}$ neurons infected by Chr2 were about 29.8% of total LPAG cells (**Supplementary Fig. 8c**). About 87.4% of $Vgat^+$ neurons by Chr2 infection were localized to the LPAG (**Supplementary Fig. 8d**). According to the previous study²¹, 473-nm laser propagates downward in biological tissue and transmission of light power decreases exponentially with the increase of brain tissue depth. The superior colliculus is above the optical fiber, while the DR is far away from the optical fiber. Even a small amount of the virus spreading to these brain regions did not affect the results.

These data are presented in the new **Supplementary Fig. 8**.

a, Scheme for photoactivation of $LPAG^{Vgat}$ neurons. **b**, Histology of injection sites and fiber traces in the LPAG. Scale bars, 300 μm and 100 μm . **c**, Percentage of $LPAG^{Vgat-Chr2}/LPAG^{DAPI}$ neurons ($n = 4$ mice). **d**, Percentage of $Vgat^+$ neurons by Chr2 infected spread localized to the LPAG. Data are shown as the mean \pm s.e.m.

Extended Data Figure 5:

- The authors should cite a recent study by the Lin lab (Falkner et al 2020), who have shown that IPAG $VGluT2$ neurons are also active during attack action in social aggression, with simultaneous recordings of jaw EMG and IPAG spiking during attack episodes (Fig 4 of that paper).

R: In our revised manuscript, we have cited the study (Falkner et al 2020). (Line 259)

Minor comments:

- Check manuscript for grammar mistakes, e.g. line 80: replace LAG with LPAG, line 91: replace shuffled with shuffling, line 92: similar instead of approximate? line 121: similar LPAG neurons – a similar percentage of LPAG neurons ...

R: We have checked and revised our manuscript.

Reviewer #3 (Remarks to the Author):

In this manuscript by Yu and Li, et al., the authors investigate the encoding of predatory behavioral sequences in GABAergic and glutamatergic lateral periaqueductal gray neurons. The authors isolated distinct encoding of the behavioral sequences by LPAG-GABA neuron subpopulations during prey detection, chase, and attack, and the contribution of LPAG-glutamate neurons to prey attack. The authors further demonstrate the necessity of these distinct populations to these behavioral sequences using temporally resolved inhibition. Overall, the manuscript is a technical achievement, integrating neural physiology, behavior, and circuit function and represents a significant advance in understanding the neural circuit dynamics underlying predatory behavior. I have only a few suggestions for strengthening the overall manuscript.

Specific comments:

1. The authors nicely demonstrate that inhibition of LPAG-Vglut2 neurons does not affect free reward consumption. However, their analysis of this population is much more superficial than the analysis of the LPAG-Vgat population. A more thorough analysis to juxtapose to the LPAG-Vgat population is warranted.

R: Thanks for this constructive suggestion. To test whether activating LPAG^{Vglut2} neurons would affect hunting behavior, we unilaterally injected AAV5-EF1a DIO-hChR2 (H134R)-EYFP vectors into the LPAG of Vglut2-IRES-Cre mice and implanted optical fibers above. We found that photoactivation of LPAG^{Vglut2} neurons with viral spread over the DLPAG (**Supplementary Fig. 9a**) induced high-velocity movement, similar to the flight behavior in active defensive responses reported before¹² (**Supplementary Fig. 9b**). However, photoactivation of LPAG^{Vglut2} neurons induced freezing-like passive defensive behavior as reported before¹² (**Supplementary Fig. 9c-e**). These data show that LPAG^{Vglut2} neurons by themselves or plus the adjacent area strongly affect defensive responses and movement, which presumably masks other behaviors such as hunting. Therefore, we did not test the hunting effect after the photoactivation. Instead, we used photoinhibition approaches to test their function in hunting (**Fig.4**)

These data are presented in the new **Supplementary Fig. 9**.

a, Left, scheme of virus injection and fiber implantation. Right, histology of injection site and fiber trace in the DL/LPAG. Scale bar, 300 μm . **b**, Photoactivation of DL/LPAG^{Vglut2} neurons induced flight responses ($n = 4$ mice for each group). **c**, Scheme of photoactivation of LPAG^{Vglut2} neurons. **d**, Z-score of velocity in ChR2 and EYFP groups. **e**, Photoactivation of LPAG^{Vglut2} neurons increased immobility duration ($***P < 0.001$, Mann Whitney U test). Data are shown as the mean \pm s.e.m.

2. It is not clear why the data in figure 6 is normalized to the EYFP control. The authors should provide the EYFP data and do conventional statistical analysis of the input population inactivation compared to the controls and between groups.

R: Thanks for the advice. We have performed conventional statistical analysis (two-way ANOVA, Tukey's multiple comparisons test) to address how individual inputs influence the distinct predatory phase. It did not change our conclusion. Basically, we found that there was a more substantial effect on prolonging the latency to attack after LH lesions as compared to CeA/ZI lesions (**Fig. 6b**). The parameters representing the chase and attack, including the chase numbers, attack segments and capture duration after CeA lesions, were more severely impaired than those after LH/ZI lesions (**Fig. 6c, e, f**). These data were consistent with the role of these LPAG-projecting GABAergic neurons from the CeA in mediating pursuit and the biting behavior during hunting²². We did not find a significant difference in the attack efficiency after ablation of LPAG-projecting GABAergic neurons from the CeA/LH/ZI (**Fig. 6d**). The eating duration was not significantly different between these lesions (**Fig. 6g**). Together, these data conclude that removing either of the CeA/LH/ZI inputs to the LPAG is sufficient to interrupt all hunting motor actions except eating, including prey detection, chase, attack, and capture. However, the extents of interruptions after removing these inputs are different. Removing ZI inputs has a more substantial effect on the movement speed (**Supplementary Fig. 14k, l**), removing LH inputs has a more substantial effect on latency to attack, while removing CeA inputs has more profound effect on prey chase and attack.

These data are presented in the new **Fig. 6**.

a, Experimental scheme of ablation of LPAG-projecting GABAergic neurons from the CeA/LH/ZI. **b**, Comparison of the effect on the latency to attack after ablation of LPAG-projecting GABAergic neurons from the CeA/LH/ZI ($n = 30$ mice, two-way ANOVA, Tukey's multiple comparisons test, $***P < 0.001$, $**P < 0.01$, $P > 0.05$, n.s., no significance). **c**, Comparison of the effect on the chase rounds after ablation of LPAG-projecting GABAergic neurons from the CeA/LH/ZI ($n = 30$ mice, two-way ANOVA, Tukey's multiple comparisons test, $*P < 0.05$, $P > 0.05$, n.s., no significance). **d**, Comparison of the effect on the attack efficiency after ablation of LPAG-projecting GABAergic neurons from the CeA/LH/ZI ($n = 30$ mice, two-way ANOVA, Tukey's multiple comparisons test, $P > 0.05$, n.s., no significance). **e**, Comparison of the effect on the attack segments after ablation of LPAG-projecting GABAergic neurons from the CeA/LH/ZI ($n = 30$ mice, two-way ANOVA, Tukey's multiple comparisons test, $**P < 0.01$, $*P < 0.05$, $P > 0.05$, n.s., no significance). **f**, Comparison of the effect on the capture duration after ablation of LPAG-projecting GABAergic neurons from the CeA/LH/ZI ($n = 30$ mice, two-way ANOVA, Tukey's multiple comparisons test, $****P < 0.0001$, $P > 0.05$, n.s., no significance). **g**, Comparison of the effect on the eating duration after ablation of LPAG-projecting GABAergic neurons from the CeA/LH/ZI ($n = 30$ mice, two-way ANOVA, Tukey's multiple comparisons test, $P > 0.05$, n.s., no significance). **h**, Summary of functions of LPAG across predatory hunting. A neuronal ensemble sequence is formed in the LPAG across the whole predatory process. Vgat⁺ population (light purple) in the LPAG encodes the introduction, chase and attack, while Vglut2⁺ population (light blue) encodes the attack. The neuronal activity of LPAG clusters across predatory processes depends on GABAergic inputs from the CeA (green line), LH (orange line) and ZI (purple line). GABAergic inputs from the CeA mainly control prey chase and attack, GABAergic inputs from the LH control prey attack, and GABAergic inputs from the ZI mainly control introduction and chase phases. The role of the inputs from the MPA on hunting needs further investigation. Data are shown as the mean \pm s.e.m.

3. The circuit level analysis represented in Figure 6h is incomplete. The authors do not investigate how the different inputs to the LPAG regulate the LPAG-Vglut2 population. It is not necessary to provide a full analysis of the role of these inputs in regulating the in vivo firing properties of LPAG-Vglut2 neurons during predatory behavior; however, the authors should perform optogenetic analysis of connectivity using slice electrophysiology. This can be easily achieved injecting AAV-FLEX-ChR2-EYFP into the CeA, LH, or ZI of Vgat-Cre mice and AAV-FLEX-mCherry into the LPAG. This will isolate GABA neurons in the LPAG, and the authors can record labeled and unlabeled neurons (presumably glutamate) neurons.

R: Thanks for this advice and we performed the suggested experiment. To assess functional connectivity of CeA/LH/ZI-LPAG pathway, we unilaterally injected AAV-hSyn-FLEX-EGFP-WPRE-hGHpA into the LPAG of Vgat-IRES-Cre mice and injected AAV-hEF1a-DIO-hChR2 (H134R)-mCherry into the CeA, the LH, and the ZI, and recorded postsynaptic currents from Vgat⁺ and Vgat⁻ (presumably glutamate) neurons in the LPAG by optogenetically activating CeA/LH/ZI-LPAG projections, respectively. We found that Vgat⁺ and Vgat⁻ neurons recorded in the LPAG received monosynaptic inhibitory inputs from the CeA, the LH and the ZI. Also, 42.9% of CeA^{Vgat}-projecting LPAG neurons were GABAergic and 30% of CeA^{Vgat}-projecting LPAG neurons were glutamatergic, 61.1% LH^{Vgat}-projecting LPAG neurons were GABAergic and 40% of LH^{Vgat}-projecting LPAG neurons were glutamatergic, while 42.9% ZI^{Vgat}-projecting LPAG neurons were GABAergic and 40.9% of ZI^{Vgat}-projecting LPAG neurons were glutamatergic. Altogether, these data indicate that the CeA, the LH, and the ZI are functionally connected with the LPAG, respectively.

These data are presented in the new **Supplementary Fig. 11**.

a-c, Connectivity of CeA-LPAG pathway. **a**, Experimental scheme. **b**, Example traces (left, by application of PTX) and quantitative analyses (right) of the amplitude of light-evoked IPSCs from the LPAG^{Vgat+} or LPAG^{Vgat-} neuron (two-tailed unpaired t-test, $P > 0.05$, n.s., no significance). **c**, Example traces (left, by application of TTX and 4-AP) and connection probabilities (right) of CeA-LPAG pathway. **d-f**, Connectivity of LH-LPAG pathway. **d**, Experimental scheme. **e**, Example traces (left, by application of PTX) and quantitative analyses (right) of the amplitude of light-evoked IPSCs from the LPAG^{Vgat+} or LPAG^{Vgat-} neuron (two-tailed unpaired t-test, $P > 0.05$, n.s., no significance). **f**, Example traces (left, by application of TTX and 4-AP) and connection probabilities (right) of LH-LPAG pathway. **g-i**, Connectivity of ZI-LPAG pathway. **g**, Experimental scheme. **h**, Example traces (left, by application of PTX) and quantitative analyses (right) of the amplitude of light-evoked IPSCs from the LPAG^{Vgat+} or LPAG^{Vgat-} neuron (two-tailed unpaired t-test, $P > 0.05$, n.s., no significance). **i**, Example traces (left, by application of TTX and 4-AP) and connection probabilities (right) of ZI-LPAG pathway. **j**, Connection probabilities of CeA/LH/ZI-LPAG pathway. Data are shown as the mean \pm s.e.m.

4. The resolution of how GABA inputs to the LPAG regulate the activation of GABA neurons during hunting remains unclear. The authors propose disinhibitory mechanisms, but this is relatively unsatisfying. The authors should provide an analysis of the projection density and anatomical resolution of the different inputs to the LPAG to establish if there is any anatomical segregation of these inputs at the level of the LPAG.

R: Thanks a lot for the suggestion. Firstly, we analyzed the projection density of different inputs using slice electrophysiology as stated above. We found that 42.9% of CeA^{Vgat}-projecting LPAG neurons were GABAergic and 30% of CeA^{Vgat}-projecting LPAG neurons were glutamatergic, 61.1% LH^{Vgat}-projecting LPAG neurons were GABAergic and 40% of LH^{Vgat}-projecting LPAG neurons were glutamatergic, while 42.9% ZI^{Vgat}-projecting LPAG neurons were GABAergic and 40.9% of ZI^{Vgat}-projecting LPAG neurons were glutamatergic.

These data are reported in the new **Supplementary Fig. 11j**.

j, Connection probabilities of CeA/LH/ZI-LPAG pathway.

Secondly, to anatomically quantify LPAG-projecting CeA/LH/ZI neurons, we injected AAV2/9-hEF1a-fDIO-EYFP and AAV2/8-hSyn-DIO-mCherry into the LPAG, simultaneously injected AAV2/1-hSyn-Cre and AAV2/1-hSyn-Flp into any two brain regions (**Supplementary Fig. 12a-c**). Interestingly, we found that only a small percentage of overlap between any of the two types of projecting neurons, including the CeA and LH (**Supplementary Fig. 12d, g**), the CeA and ZI (**Supplementary Fig. 12e, h**), and the LH and ZI (**Supplementary Fig. 12f, i**), respectively. Also, we quantified the cell numbers at different bregma sites of the LPAG and did not find an apparent difference in bregma sites (**Supplementary Fig. 12j**). We did not directly perform the analysis of LPAG neurons simultaneously receive three types of projections from the CeA, the ZI and LH. However, based on the results in Supplementary Fig. 12d-i, this type of LPAG neurons would be few. Altogether, these data indicate that the CeA, the LH, and the ZI are anatomically connected with distinct subpopulations of LPAG neurons, respectively.

These data are presented in the new **Supplementary Fig. 12**.

a-c, Experimental scheme of quantifying LPAG-projecting CeA/LH/ZI neurons. **d**, Top, images of LPAG-projecting CeA (red)/LH (green) neurons in WT mice. Scale bars, 100 μ m and 50 μ m. Bottom, the overlap between LPAG-projecting CeA neurons (red) and LPAG-projecting LH neurons (green) ($n = 6$ mice). **e**, Top, images of LPAG-projecting CeA (red)/ZI (green) neurons in WT mice. Scale bars, 100 μ m and 50 μ m. Bottom, the overlap between LPAG-projecting CeA (red) neurons and LPAG-projecting ZI (green) neurons ($n = 6$ mice). **f**, Top, images of LPAG-projecting LH (green)/ZI (red) neurons in WT mice. Scale bars, 100 μ m and 50 μ m. Bottom, the overlap between LPAG-projecting LH neurons (green) and LPAG-projecting ZI (red) neurons ($n = 6$ mice). The bregma in **d-f** was -3.52 mm. **g**, Top, images of LPAG-projecting CeA (red)/LH (green) neurons in WT mice. Scale bars, 100 μ m and 50 μ m. Bottom, the overlap between LPAG-projecting CeA neurons (red) and LPAG-projecting LH neurons (green) ($n = 6$ mice). **h**, Top, images of LPAG-projecting CeA (red)/ZI (green) neurons in WT mice. Scale bars, 100 μ m and 50 μ m. Bottom, the overlap between LPAG-projecting CeA (red) neurons and LPAG-projecting ZI (green) neurons ($n = 6$ mice). **i**, Top, images of LPAG-projecting LH (green)/ZI (red) neurons in WT mice. Scale bars, 100 μ m and 50 μ m. Bottom, the overlap between LPAG-projecting LH neurons (green) and LPAG-projecting ZI (red) neurons ($n = 6$ mice). The bregma in **g-i** was -4.04 mm. White arrows in **d-i** represented overlap neurons. **j**, Quantification of LPAG-projecting CeA/LH/ZI neurons. Data are shown as the mean \pm s.e.m.

References

- 1 Harvey, C. D., Coen, P. & Tank, D. W. Choice-specific sequences in parietal cortex during a virtual-navigation decision task. *Nature* 484, 62-68, (2012).
- 2 Wainwright, P. C., Mehta, R. S. & Higham, T. E. Stereotypy, flexibility and coordination: key concepts in behavioral functional morphology. *The Journal of experimental biology* 211, 3523-3528, (2008).
- 3 Ewert, J.-P. Neuroethology of releasing mechanisms: Prey-catching in toads. *Behavioral and brain sciences* 10, 337-405, (1987).
- 4 Mearns, D. S., Donovan, J. C., Fernandes, A. M., Semmelhack, J. L. & Baier, H. Deconstructing Hunting Behavior Reveals a Tightly Coupled Stimulus-Response Loop. *Current biology : CB* 30, 54-69 e59, (2020).
- 5 Li, Y. et al. Hypothalamic Circuits for Predation and Evasion. *Neuron* 97, 911-924 e915, (2018).
- 6 Comoli, E., Ribeiro-Barbosa, E. R., Negrao, N., Goto, M. & Canteras, N. S. Functional mapping of the prosencephalic systems involved in organizing predatory behavior in rats. *Neuroscience* 130, 1055-1067, (2005).
- 7 Pfeiffer, B. E. & Foster, D. J. Hippocampal place-cell sequences depict future paths to remembered goals. *Nature* 497, 74-79, (2013).
- 8 Grosmark, A. D. & Buzsaki, G. Diversity in neural firing dynamics supports both rigid and learned hippocampal sequences. *Science* 351, 1440-1443, (2016).
- 9 Peters, A. J., Chen, S. X. & Komiyama, T. Emergence of reproducible spatiotemporal activity during motor learning. *Nature* 510, 263-267, (2014).
- 10 Tanaka, Y. H. et al. Thalamocortical Axonal Activity in Motor Cortex Exhibits Layer-Specific Dynamics during Motor Learning. *Neuron* 100, 244-258 e212, (2018).
- 11 Gallego, J. A., Perich, M. G., Miller, L. E. & Solla, S. A. Neural Manifolds for the Control of Movement. *Neuron* 94, 978-984, (2017).
- 12 Tovote, P. et al. Midbrain circuits for defensive behaviour. *Nature* 534, 206-212, (2016).
- 13 Deng, H., Xiao, X. & Wang, Z. Periaqueductal Gray Neuronal Activities Underlie Different Aspects of Defensive Behaviors. *The Journal of neuroscience : the official journal of the Society for Neuroscience* 36, 7580-7588, (2016).
- 14 Bandler, R. & Carrive, P. Integrated defence reaction elicited by excitatory amino acid microinjection in the midbrain periaqueductal grey region of the unrestrained cat. *Brain research* 439, 95-106, (1988).
- 15 O'Connell, L. A. & Hofmann, H. A. The vertebrate mesolimbic reward system and social behavior network: a comparative synthesis. *The Journal of comparative neurology* 519, 3599-3639, (2011).
- 16 Noriuchi, M., Kikuchi, Y. & Senoo, A. The functional neuroanatomy of maternal love: mother's response to infant's attachment behaviors. *Biological psychiatry* 63, 415-423, (2008).
- 17 Bandler, R. & Shipley, M. T. Columnar organization in the midbrain periaqueductal gray: modules for emotional expression? *Trends in neurosciences* 17, 379-389, (1994).
- 18 Evans, D. A. et al. A synaptic threshold mechanism for computing escape decisions. *Nature* 558, 590-594, (2018).

- 19 Zhou, T. et al. History of winning remodels thalamo-PFC circuit to reinforce social dominance. *Science* 357, 162-168, (2017).
- 20 Zhao, Z. D. et al. Zona incerta GABAergic neurons integrate prey-related sensory signals and induce an appetitive drive to promote hunting. *Nature neuroscience* 22, 921-932, (2019).
- 21 Yizhar, O., Fenno, L. E., Davidson, T. J., Mogri, M. & Deisseroth, K. Optogenetics in neural systems. *Neuron* 71, 9-34, (2011).
- 22 Han, W. et al. Integrated Control of Predatory Hunting by the Central Nucleus of the Amygdala. *Cell* 168, 311-324 e318, (2017).

REVIEWERS' COMMENTS

Reviewer #1 (Remarks to the Author):

The authors have thoughtfully addressed all my previous concerns with additional analyses and experiments.

Reviewer #2 (Remarks to the Author):

I would like to congratulate the authors on a beautiful study, they have answered all open questions, and included important new analyses. I fully support the publication of their work in its current form.

Minor note: We seem to disagree on the topic of 'stereotypy' and 'flexibility' of (instinctive) behaviours, so I thank the authors for trying to discuss these concepts in their discussion.

Reviewer #3 (Remarks to the Author):

The authors have adequately addressed all of my previous concerns and have significantly improved their manuscript. As per my previous assessment, I now can firmly conclude that this study represents a significant advance in understanding the neural circuit dynamics underlying predatory behavior.